

8       **Improvement of Soil Respiration Parameterization in a Dynamic Global Vegetation**
9        **Model and Its Impact on the Simulation of Terrestrial Carbon Fluxes**

14        Dongmin Kim[1], Myong-In Lee[1*], and Eunkyo Seo[1]
16       [1]School of Urban and Environmental Engineering, UNIST, Ulsan, Korea

--------------------------------------------------------------------------------------------------------------------

Corresponding author address: Dr. Myong-In Lee
School of Urban and Environmental Engineering
Ulsan National Institute of Science and Technology,
100 Banyeon-ri, Ulju-gun, Ulsan 689-798, Korea
Email: milee@unist.ac.kr





**Abstract**

Soil decomposition is one of the critical processes for maintaining a terrestrial ecosystem

and the global carbon cycle. The sensitivity of soil respiration (Rs) to temperature, the so-called
Q10 value, is required for parameterizing the soil decomposition process and is assumed to be
a constant in conventional numerical models, while realistically it is not in cases of
spatiotemporal heterogeneity. This study develops a new parameterization method for
determining Q10 by considering the soil respiration dependence on soil temperature and
moisture obtained by multiple regression. This study further investigates the impacts of the
new parameterization on the global terrestrial carbon flux. Our results show that non-uniform
spatial distribution of Q10 tends to represent the dependence of the soil respiration process on
heterogeneous surface vegetation type compared with the control simulation using a uniform
Q10. Moreover, it tends to improve the simulation of the observed relationship between soil
respiration and soil temperature and moisture, particularly over cold and dry regions. The new
parameterization improves the simulation of gross primary production (GPP). It leads to a more
realistic spatial distribution of GPP, particularly over high latitudes (60–80 N) where the
original model has a significant underestimation bias. In addition, overestimation bias of GPP
in the tropics and the midlatitudes is significantly reduced. Improvement in the spatial
distribution of GPP leads to a substantial reduction of global mean bias of GPP from + 9.11 to
+ 1.68 GtC yr$^{-1}$ compared with the FLUXNET-MTE observation data.




## 1. Introduction


Vegetated land surface affects climate (Foley et al., 1998; Sellers et al., 1986) and is affected
by climate significantly (Bonan, 2008), forming complex interactions and feedback loops
critical to climate change (Friedlingstein et al., 2006; Gregory et al., 2009). The land surface
components of Earth System Models (ESMs) have evolved from only representing biophysical
processes (i.e., hydrology and energy cycling) to including biogeochemical processes, such as
dynamic vegetation change and carbon and nutrient cycles driven by ecosystems (Oleson et al.,
2013; Sitch et al., 2003; Wang et al., 2010). The carbon balance of terrestrial ecosystems is the
result of the balance between carbon uptake and loss by plants and soil respiration (Beer et al.,
2010; Malhi et al., 1999; Le Que re et al., 2009, 2014; Luyssaert et al., 2007; Trumbore, 2006).
Which terrestrial ecosystems act dominantly as sinks or sources has been a subject of
considerable interest in studies of future climate change. Precise evaluation for each sink and
source component and their responses to environmental factors are essential for reliable
projection of future climate change by ESMs.
Future climate change projection by various ESMs is diverse and highly uncertain
(Friedlingstein et al., 2006). One of the main causes seems to be related to our poor knowledge
on carbon exchange by soil, leading to significant diversity among the model simulations.
Diversity in the parameterization of photosynthesis at the leaf level is small compared with that
of the soil decomposition process in contemporary ESMs with an interactive carbon cycle.
Microbial decomposition of soil organic matter produces a major carbon flux from the
subsurface biosphere. A few studies suggest that global warming would be accelerated by the
release of $CO_2$ from soil (e.g., Suseela et al., 2012). However, the amplitude of the soil
decomposition process has not been quantified and is highly uncertain, mostly due to the lack
of observation data and poor estimates of it based on soil temperature (Sussela et al., 2012).
The reduction of uncertainty in the soil biogeochemical process remains a challenge for the



ESM modeling community.
Soil respiration (Rs) is considered a significant source of $CO_2$ from terrestrial ecosystems.
Recent studies suggest that $CO_2$ emission change by soil should be largely driven by surface
temperature change (Bond-Lamberty and Thomson, 2010). At global, regional and local scales,
soil temperature and soil moisture are considered the most important abiotic parameters
determining Rs (Kutsch et al., 2009). Empirical response functions based on heterogeneous
field measurements are commonly used to derive annual estimates of Rs (Tang et al., 2005).
The sensitivity of soil respiration (Rs) to temperature, the so-called Q10 value, is required
for parameterizing the soil decomposition process. Despite a lack of observation data from field
studies for Rs and its dependence on soil temperature, some previous studies have suggested
that the Q10 value derived from soil respiration measurement tends to decrease with
temperature because substrate availability decreases as temperature increases (Belay-Tedla et
al., 2009). All the abiotic and biotic factors such as soil temperature (Lloyd and Taylor, 1994;
Kirschbaum, 1995; Luo et al., 2001), moisture (Davidson et al., 1998; Reichstein et al., 2002;
Hui and Luo, 2004), and soil organic matter (Taylor et al., 1989; Liski et al., 1999; Wan and
Luo, 2003) are heterogeneous, showing substantial spatial variation globally. Accordingly,
estimated Q10 from measured soil respiration possibly varies at various geographic locations
(Xu and Qi, 2001).
Based on the aforementioned studies, Zhao et al. (2009) developed an inverse model to
retrieve the global pattern of heterogeneous Q10 values by assimilating soil organic carbon
data with a process-based biogeochemical model. They suggested that spatial distribution of
Q10 values changes according to vegetation type, with an increasing tendency as latitude
increases. The impact on the estimation of carbon release due to Q10 variation in space is a
significant change of approximately 25–40 % compared with the use of a constant Q10 value
in Zhao et al. (2009). This result suggests that the determination of Q10 value is very important





for the simulation of carbon-climate feedback and future climate change. However, most
advanced ESMs that participated in Coupled Model Intercomparison Project Phase 5 (CMIP5)
still use a globally constant Q10 value in the dynamic global vegetation model. In this case, the
sensitivity of subsurface carbon flux under global warming condition would not be reflected in
the model simulation.

Motivated by the above, this study developed a new parameterization method for

determining Q10 by considering the dependence of soil respiration on soil temperature and
moisture, the relationship of which was obtained from multiple regression with those two
predictors. The variation of dominant vegetation type for the given area was also considered
when determining Q10. Community Land Model version 4 (CLM4) has the parameterization
of the interactive carbon and nitrogen (C-N) cycle for the dynamic vegetation model, which
was used to derive realistic spatial distributions of Q10. This study further investigates the
impacts of the new parameterization on the global carbon cycle.

Section 2 describes the observation and modeling data used in this study and the modeling

method used to obtain the distribution of Q10. Section 3 provides the results from the off-line
dynamic vegetation model test with prescribed atmospheric states. In addition, the results from
a fully interactive ESM model test with the modified Q10 are provided in that section.
Summary and further discussion are provided in Section 4.

**2. Data, Methods, and Experiments**

**2.1. Data**

FLUXNET-MTE (Multi Tree Ensemble) data (Jung et al., 2009) is used to validate GPP.
FLUXNET provides the global distribution of carbon and water fluxes in the vegetated land
surface and its temporal variation, which were derived from upscaling eddy covariance
measurements at the flux tower sites using a statistical machine-learning algorithm. The data





provide the information on terrestrial carbon and water cycles globally (Jung et al. 2009). The
data's spatial resolution is 0.5° X 0.5° (lat./lon.) and is monthly for 23 years (1983–2009).
This study also used the Moderate Resolution Imaging Spectroradiometer (MODIS) GPP and
net primary production (NPP) data. Autotrophic respiration (Ra) by plants was determined by
subtracting NPP from GPP by definition. This study used the gridded data for the global
domain at 0.5° X 0.5°  horizontal resolution. These data are originally from MODIS17A3 GPP
and NPP products in HDF EOS (Hierarchical Data Format – Earth Observing System) format
with a native resolution of 1 km (Running et al., 2004). Each tile is 1200 X 1200 km (Zhao et
al., 2005).
When GPP is compared between in situ observation-based FLUXNET-MTE and satellite-
based MODIS, the two datasets show a minor difference for the overlapping period (2000–
2006). The global GPP of FLUXNET-MTE is 101.13 GtC yr$^{-1}$ and that of MODIS is 100.51
GtC yr$^{-1}$, which is less than 1 % of the total value.
Soil respiration (Rs) was verified using the data from Hashimoto et al. (2015). The data were
also used for the parameterization of soil respiration (described in detail in Section 2.2).
Although only directly observed soil respiration is available from SRDB data version 3 (Bond-
Lamberty and Thomson, 2010), it has limited sampling for boreal cold regions (i.e., tundra and
northern Siberian) as well as unpopulated regions in the tropics, covering a significant portion
of the global biosphere. The data from Hashimoto et al. were derived using SRDB data and the
empirical soil respiration model with specified climate conditions for surface air temperature
and precipitation. The model was modified and updated from the original version of Raich et
al. (2002). Global land use data in a synergetic land cover product (SYNMAP, Jung et al., 2006)
using a Bayesian calibration scheme were used to determine the best parameter set for
assuming the climate-driven model of soil respiration. The climate-forcing data were obtained
from CRU version 3.21 climate data (University of East Anglia Climatic Research Unit, 2013).





These data was applied monthly at a spatial resolution of 0.5° X 0.5° (lat./lon.).
All the data were regridded onto 1.9° X 2.5° lat./lon. grids for comparison with the CLM4
simulation at this resolution.

**2.2. Q10 Parameterization**
Most dynamic vegetation models implemented in current ESMs, including CLM4, adopt a
simple type of empirical equation for Rs, which is proportional to the soil decomposition flux
of carbon at the root zone. The decomposition flux is calculated by multiplying the carbon
amount from dead leaf by the rate scalar ($R_{scalar}$), representing the effects of the physical
environmental condition such as soil temperature ($T_{scalar}$) and moisture ($W_{scalar}$) as:

$R_{scalar} = T_{scalar} * W_{scalar}$ ,            (1)

where $T_{scalar}$ is basically an exponential function of temperature from van't Hoff (1898). It is
implemented in CLM4 as in the following equation:

$T_{scalar} = Q_{10}^{[\frac{T_j - T_{ref}}{10}]}$,            (2)

where $T_j$ is the temperature at the $j$-th soil level, and $T_{ref}$ is the reference temperature of 25
°C. CLM4 considers temperature for the top 5 soil levels as representing the root zone (approx.
29 cm depth). $Q_{10}$ is specified as a constant value of 1.5 in the standard CLM4 model. The
moisture scalar ($W_{scalar}$) is based on Andren and Paustian (1987), which describes the potential
for soil water decomposition as

$W_{scalar} = \sum_{j=1}^{5} \frac{\log(\frac{\Psi_{min}}{\Psi_j})}{\log(\frac{\Psi_{min}}{\Psi_{max}})}$   ,      (3)

where $\Psi_j$ is the soil water potential at the level $j$ defined from the exponential of volumetric





soil moisture ($m^3$ $m^{-3}$). $\Psi_{max}$ is the maximum potential depending on soil type, and $\Psi_{min}$ is
the minimum value of -10 MPa, regardless of soil type. The range of $W_{scalar}$ is 0 to 1 by
setting to 0 when the $\Psi_j$ is below $\Psi_{min}$, and setting to 1 when $\Psi_j$ is above $\Psi_{max}$.

For improving the $Rs$ parameterization in CLM4, this study considers a spatiotemporal

change of $Q_{10}$ in (2). We developed a multiple regression model for $Q_{10}$ based on Qi et al.
[2002], which assumes that the rate of $R_s$ change depends entirely on soil temperature ($T$) and
soil moisture ($M$). These two physical variables are well-known important factors for soil
biological processes. The fractional instantaneous change of $R_S$ by soil temperature $q$ is defined
as
$$q(T, M) = \frac{1}{R_s} \frac{dR_s}{dT} \qquad (4)$$

$Q_{10}$ is defined as the relative change of $R_s$ at a temperature increase of 10 degrees, which

can be described in the following equations:
$$Q_{10}(T, M) = \frac{R_s(T+5, X)}{R_s(T-5, X)} \qquad , \qquad (5)$$
$$Q_{10} = e \int_{T-5}^{T+5} q(T, X)\ dT \qquad , \qquad (6)$$
where $X$ is any additional independent variable to predict Rs. In this case, only soil moisture
($M$) is considered. From (6), $Q_{10}$ is a monotonic function of q, and the factor affecting q also
influences $Q_{10}$. Therefore, the change of Rs is decomposed into the change by temperature
and the change by moisture:
$$\frac{dR_s}{dT} = \frac{\partial R_s(T, M)}{\partial M} \frac{dM}{dT} + \frac{\partial R_s(T, M)}{\partial T} \qquad . \qquad (7)$$

Inserting (7) into (4), the equation for q is rewritten as

$$q(T, M) = \frac{1}{R_s} \left[ \frac{\partial R_s(T, M)}{\partial M} \frac{dM}{dT} + \frac{\partial R_s(T, M)}{\partial T} \right] \qquad , \qquad (8)$$



where $dM/dT$ = -1/2.2 = -0.455, as suggested by Xu and Qi (2001).
Through a multiple regression analysis, the relationships between $R_s$ and T and between
$R_s$ and M were obtained. In this study, multiple regression was conducted for each plant
function type (PFT) for 16 classifications in CLM4. The $Q_{10}$ multiple regression model
developed in this study has an advantage over the treatment of constant value in the standard
CLM4 model. First, the dependence of Rs on soil moisture and temperature can be dependent
on PFT. In addition, this approach is able to consider the nonlinear relationship between $R_s$
and the two major environmental variables of soil temperature and moisture, supported by
recent observational studies (Davidson et al., 1998; Raich et al., 2002).
Crucial for the parameterization of $Q_{10}$ are the quality of the reference data and the degree
of fitting for multiple regression. The observation data for $R_S$ were obtained from Hashimoto
et al. (2015) soil respiration data. The parametrization requires the dependence of soil
respiration on subsurface temperature and moisture; these data are also not available from in
situ observations. To obtain these variables, this study conducted a land surface reanalysis for
the most recent 30 years (1981–2010), using the off-line land-surface model driven by observed
meteorological forcing data from Sheffield et al. (2006). The forcing data by Sheffield et al.
consist of the observation-based datasets of precipitation, air temperature, and radiation. The
Global Precipitation Climatology Project (GPCP; Huffman et al., 2001) and the Tropical
Rainfall Measuring Mission (TRMM: Huffman et al. 2003) 3B42RT were utilized for the
rescaling of daily and 3-hour precipitation, respectively. The surface temperature observation
is based on the Climatic Research Unit (CRU) 2.0 product (Mitchell et al. 2004). The radiation
data was based on the NASA Langley monthly surface radiation budget (Stackhouse et al.,
2004). Remaining meteorological conditions, such as surface wind and humidity, were from
the National Centers for Environmental Prediction–National Center for Atmospheric Research
(NCEP-NCAR) atmospheric reanalysis. The offline land-surface model integration was



conducted with 3-hour forcing data. Our detailed procedure of the off-line land-surface model
integration is also found in Seo et al. (2016, in manuscript).

Figure 1 shows the r-squared values from the multiple regression for soil respiration for

various PFT types. In most vegetation types, the regression by soil temperature and moisture
tends to exhibit high values close to 1. The regression results are better than they are when the
multi-model ensemble average of soil temperature and moisture from 13 Global Soil Wetness
Project (GSWP2) land surface model outputs (Dirmeyer et al., 2006) were applied to the
multiple regression. This difference is attributed mostly to a better quality of forcing data by
Sheffield et al. (2006), such as the use of daily precipitation data instead of monthly values in
GSWP2 and a longer training period from 1983–2010 than was used for GSWP2 data (1986–
1995). The r-square value was found to be comparable when the period of forcing data was
reduced.

**2.3. Experiments**

Two sets of off-line CLM4 simulations were conducted with identical meteorological forcing

for 23 years (1983–2005), where the only difference was the specification of Q10 in the control
run (CTL) and the state-dependent Q10 in every time interval (EXP). Figure 2 shows the time
average of Q10 values, where the geographical change is clear according to the dominant PFTs
and climate conditions. Generally, the regions of lower canopy plants with cold soil
temperatures exhibit relatively higher values, significantly higher than the default value of 1.5
in CTL. In contrast, the regions of lower Q10 values are located at low latitudes in high
temperatures, such as the Amazon and the Maritime Continent. This result suggests that soil
respiration is more sensitive to the change of soil temperature in boreal vegetated regions in
cold climates.

The time average of the off-line simulation from the standard run (CTL) in Figure 3 is very



similar to the fully interactive integration of the same model in terms of the spatial bias pattern
for the terrestrial carbon fluxes, presumably inherited by the deficiencies in the parametrization
of the dynamic vegetation model. Both simulations tend to overestimate GPP over the tropics
and underestimate it in high latitudes. The bias pattern of Rs is also quite similar with no
significant difference. Despite the fact that in the simulated climatic condition the fully
interactive run should be different from the observation used to drive the off-line CLM4, much
resemblance in the terrestrial carbon–flux bias pattern suggests that the deficiency in the
dynamic vegetation model is overwhelming the bias rather than that systematic error is
occurring in the climate condition. Therefore, our comparison in the following sections is
mostly for the off-line simulation differences between CTL and EXP.

**3. Results**
**3.1. GPP simulations by CMIP5 ESMs**
Figure 4 compares the zonal mean distribution of GPP averaged for 23 years (1983–2005)
between FLUXNET-MTE observations and the historical emission-driven simulations by
CMIP5 ESMs. Among the models, the two ESMs from CESM-BGC and NorESM share an
identical dynamic vegetation model with an interactive C-N cycle (Bonan et al., 2011). The
global GPP simulated by the multi-model ensemble (MME) of CMIP5 ESMs is 119.28 GtC yr$^-$
$^1$, which is a slight overestimation by 18 GtC yr$^{-1}$ from the FLUXNET-MTE observation.
Overall, MME shows realistic meridional variation with large values in the tropics and small
values in high latitudes. As identified in previous studies, however, the ESMs tend to
overestimate GPP significantly in the tropics (Shao et al., 2013; Anav et al., 2013). Global GPP
simulated by the two ESMs with an interactive C-N cycle is lower than the remainder of the
ESMs (– 12 GtC yr$^{-1}$). Including typical biases of overestimation of GPP over tropical belts
(20S–20N), the two models show the other GPP bias from the remainder of the ESMs, which



tend to significantly underestimate GPP, even more so than other ESMs in the Northern
Hemisphere high latitudes (> 60 N). These systematic biases are a common problem in the C-
N coupled models based on CLM4 (Bonan et al., 2011; Thornton et al., 2009). These two are
major GPP regions, where the model biases are crucial to the uncertainty of the future climate
change projections.

**3.2. Sensitivity to the Q10 parametrization**
Figure 5 shows the spatial distribution of GPP from the observations and the offline
control CLM4 simulation with the standard model (CTL). Although the simulated pattern of
GPP is generally consistent with the FLUXNET observation, the model also exhibits the
systematic biases clearly, such as significant overestimation of GPP in the tropical belts. Figure
5 also compares the Rs pattern between the observation and the offline model simulation. The
simulated pattern also shows a general agreement with the observation, such as large soil
respiration in warm and wet regions in low latitudes and less respiration in cold and dry regions
in high latitudes. However, the simulation bias in CTL shows the uniform pattern of
underestimation in almost every region except central China. This bias suggests that the
parameterization of internal soil biological processes could be misrepresented in the standard
CLM4 model.
The Rs simulation difference between CTL and EXP is given in Figure 5, in terms of
global distribution as well as zonally-averaged distribution. Overall, the modification to Q10
tends to increase Rs in almost every region. This is an improvement from CTL, although the
model now tends to overestimate Rs in some specific regions, such as the tropics and the high
latitudes in the Northern Hemisphere, such as southern Siberia, Alaska and China. Overall the
increase of Rs can be attributed to the increase of Q10 in most of the vegetated regions (Figure
2) from the standard value of 1.5. Despite the increase of Rs in EXP, the underestimation



persists over the Amazon and other large biomass regions.

The Q10 parametrization tends to enhance the relationship between Rs and soil

temperature from CTL (Figure 6). First, simulated Rs by EXP is larger than that of CTL in most
PFTs, and the difference between EXP and CTL increases with temperature. This sensitivity
also depends on the surface vegetation type. The temperature sensitivity is particularly strong
in boreal and tropical plant types. This relationship is rather unclear in the temperate plant type,
which seems to suggest the role of soil moisture in this vegetation type. The samples for the
grass type are too small to detect the sensitivity. The different sensitivities should be linked to
changes in Q10 values in EXP from CTL (Table 1). The Q10 value has been increased for the
boreal forest and boreal shrubs in EXP; whereas, it has been decreased or has no significant
change in the temperate forest type. This result is consistent with the parameterization for Q10
in Equation (2).

Figure 7 compares the GPP bias patterns in CTL and EXP. CTL shows significant biases

when sign and magnitude differ geographically. Among the biases, overestimation in the
tropics and underestimation in Siberia is evident. Although the spatial structure of bias seems
to be quite similar, implying intrinsic model deficiencies other than Q10, EXP shows an
improvement by reducing biases such as overestimation in southern Asia and China and
underestimation in northern Eurasia in CTL. However, underestimation biases in the central
part of North America and the Amazon are even larger in EXP. This change of spatial
distribution of GPP is associated with sensitivity of Rs and soil temperature. Degradation of
GPP simulation over Europe and North America is driven by the temperate plant type where
the temperature sensitivity of Rs tends to decrease (Figure 6). On the other hand, the northern
Eurasian and Chinese regions that have good improvement of GPP bias in EXP show an
enhanced relationship between Rs and temperature. This result indicates that the change of Rs
to soil temperature by Q10 variation affects not only the change in respiration but also the





carbon production (GPP) flux.
The improvement in the GPP simulation by the Q10 parameterization is illustrated better
in Figure 8, which compares the regionally averaged GPP over major 4 regions. In the global
average, EXP reduces the overestimation bias by approximately 10 % (103.81 GtC/yr)
compared with CTL simulation (111.24 GtC/yr). Little plant cover over SH (60S-20S) leads
to a smaller contribution of global GPP. The improvement over this region is not clear in EXP.
However, the overestimation bias in the tropical regions has been improved significantly. This
result is caused by the suppression of the GPP amount in the Amazon region in EXP. This
underestimation of GPP over the Amazon induces the improvement of the zonal mean
terrestrial carbon budget in EXP. The middle latitude region (20N–60N), which is dominated
by temperate forest and crop fields, also has a reduced overestimation of GPP bias compared
with CTL. In addition, simulated GPP over high latitude regions (> 60N) were improved in
EXP. Those were also the common areas of bias in the interactive C-N coupled ESM run.
The modification to the soil process parameterization can affect the rest of the terrestrial
carbon cycle by changing the carbon pool in the soil system for plant assimilation. For detailed
investigation of the impact of the Q10 parameterization, this study further investigates the
changes in the simulated terrestrial carbon cycle of each vegetation type. Figure 9 compares
the observation and the simulations using two offline runs for GPP, autotropic respiration by
plants (Ra), and Rs depending on the primary vegetation type. For the comparisons of GPP and
Ra, satellite-based MODIS data were used as the data separated GPP and Ra over vegetation
areas. In the MODIS observations, the terrestrial carbon cycle is largely contributed to by
vegetation response in tropical and temperate tree regions. Vegetation types with a short canopy
height and trees with deciduous leaves contribute less in terms of absolute amount of carbon
fluxes, although their relative changes are not trivial. Both CTL and EXP runs capture these
observed differences in the magnitude of carbon fluxes realistically. Regarding the simulation



of GPP, EXP tends to reduce the biases, particularly in temperate, tropical and crop zones. EXP
also improves the simulation of Ra in those regions. The improvement is most evident in Rs,
where the simulated values are close to the observed values in most vegetation types. Rs by
EXP has been increased in every type of vegetation from CTL, reaching values closer to the
reference observation data. According to this result, although the absolute magnitude of Rs is
much smaller compared with that of GPP and Ra, the modification of Rs by the Q10
parameterization affects the entire terrestrial carbon cycle and improves their simulations.

**4. Summary and Concluding Remarks**
Soil respiration is a crucial process in maintaining terrestrial carbon cycles. Although its
sensitivity to the physical environmental conditions such as soil temperature and moisture
depends on the type of vegetation, as supported by observational data, most contemporary
ESMs do not consider this dependence. These models thereby underestimate the effects of and
feedbacks from soil respiration on terrestrial carbon cycles. Using the CLM4 land surface
model with the interactive C-N cycle, this study developed a new parameterization method to
consider the spatiotemporal variation of Q10 that represents the sensitivity of soil respiration
to the temperature change for each different vegetation type. This sensitivity has been treated
as constant with a uniform value regardless of plant type in the original CLM4 model.
The new parameterization changes the simulation of soil respiration and the rest of
terrestrial carbon fluxes significantly by enhancing the feedback to the plant production process.
The new parameterization calculates Q10 at every time interval for each location, and this state-
dependent prescription induces the overall increase of soil respiration in most locations and
most vegetation types, improving spatially uniform negative bias in the original CLM4
simulation with constant Q10 value. The simulated sensitivity of soil respiration to soil
temperature and moisture by the new method showed more realistic features, particularly in



the temperate and cold regions. This changed soil carbon fluxes at the subsurface and affected
the simulation of GPP, where the simulation of spatial distribution of GPP has been improved
particularly over high latitudes with short canopy heights and over the tropics and warm regions,
including southern Asia and China. The improved GPP simulation over cold regions was
mostly attributed to the increase in carbon decomposition in those regions. Due to the
advancement of both respiration and primary production, carbon balance between subsurface
and surface ecosystems with soil organic matter and plants were also improved by the new Q10
parameterization. The observed ratio of soil respiration to GPP was represented better in the
new simulation, which clearly shows the dependence on the vegetation type.
The major findings from this study suggest that the modification of subsurface terrestrial
carbon cycle processes is important for improving the simulation of terrestrial carbon fluxes.
The parameterization of the photosynthetic process is still a major term crucially related to
primary production (Bonan et al. 2010; Bonan et al., 2011). Previous studies have suggested
that the improvement of canopy processes in the photosynthetic parameter in CLM4 was able
to improve the simulation by reducing the overestimation of GPP in the tropics. Despite the
improvement in the photosynthetic process in their models, respiration processes by plants and
soil are still largely uncertain due to a lack of reliable observational data and comprehensive
studies. For this reason, this study approached the modification of the soil decomposition
process, aiming to improve the terrestrial carbon cycle. In fact, the parameterization of
photosynthesis is more or less similar, with small differences in current ESMs. Still, large
uncertainties lie in the formulation of the respiration process and its parameters. This study
suggested that the improved soil decomposition process induces a change in carbon-climate
feedback intensity by changing soil respiration. In addition, the realistic description of Q10
variation in a numerical model will reduce the uncertainty of the magnitude of carbon-climate
feedback due to accurate atmospheric $CO_2$ simulation in ESMs.





**Acknowledgement**
This study is supported Basic Science Research Program through the National Research
Foundation of Korea (NRF), funded by the Ministry of Education, Science and Technology
(2012M1A2A2671851) and the Supercomputing Center/Korea Institute of Science and
Technology Information with supercomputing resources including technical support (KSC-
2015-C3-035).








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




**Table 1.** Climatological averaged Q10 values by PFTs in CLM4

|  | Temperate | Boreal | Tropical | Shrub | B. Shrub | Grass | Crop |
|---|---|---|---|---|---|---|---|
| Averaged Q10 value | 1.446 | 1.762 | 1.374 | 1.266 | 1.918 | 1.842 | 2.041 |









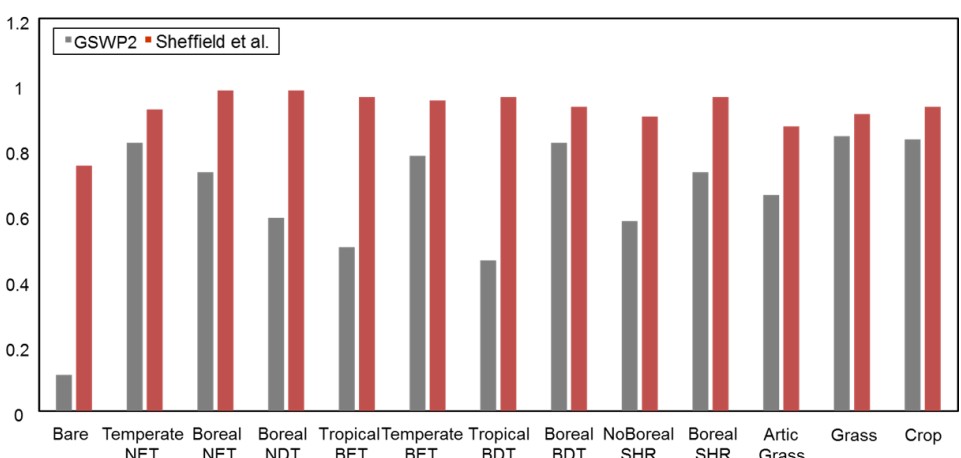


**Figure 1.** R-squared value in multiple regression by PFTs in CLM4 between soil respiration

data and soil temperature and moisture from GSWP2 multiple ensemble model data (grey bars)

and off-line model output forced by Sheffield data for 28 years (red bars).












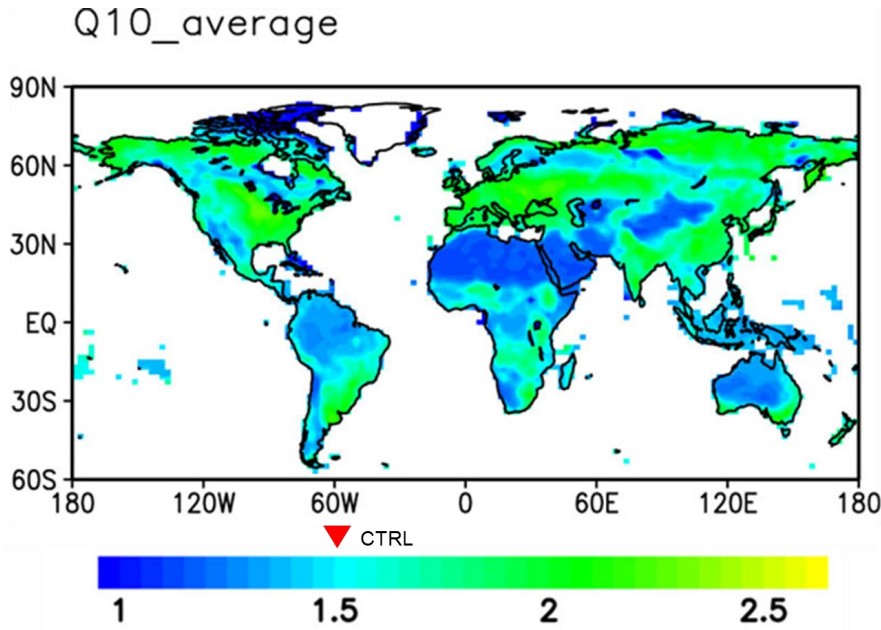


**Figure 2.** Climatological averaged Q10 spatial distribution in EXP experiment. Red filled

triangle indicates standard value of Q10 in CTL experiment.








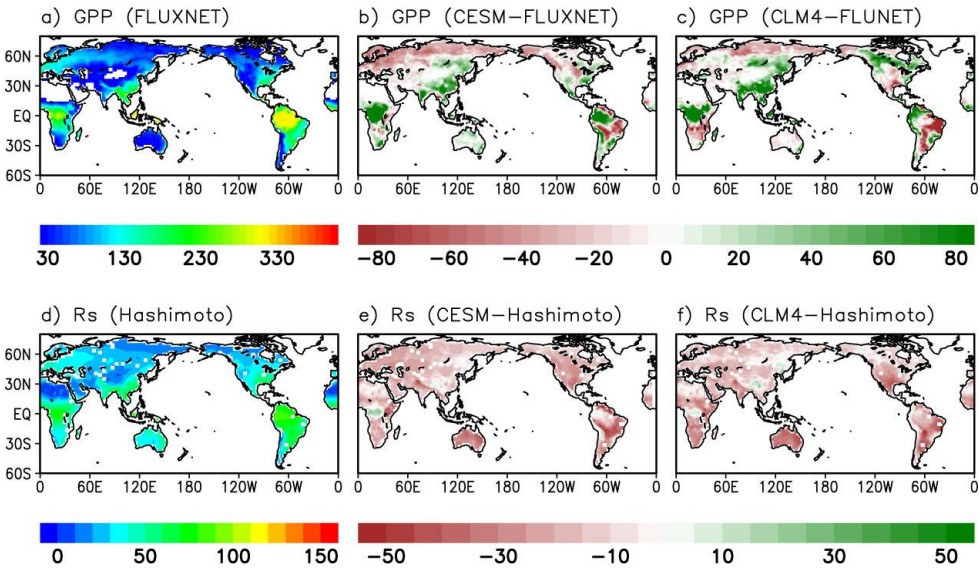

**Figure 3.** Spatial distribution of GPP(upper) and Rs (bottom) in the observation and bias

patterns of online full interactive simulation (CESM) and off-line (CLM4) experiment for 23

years (1983-2005). The unit is gC m$^2$ mon$^{-1}$.




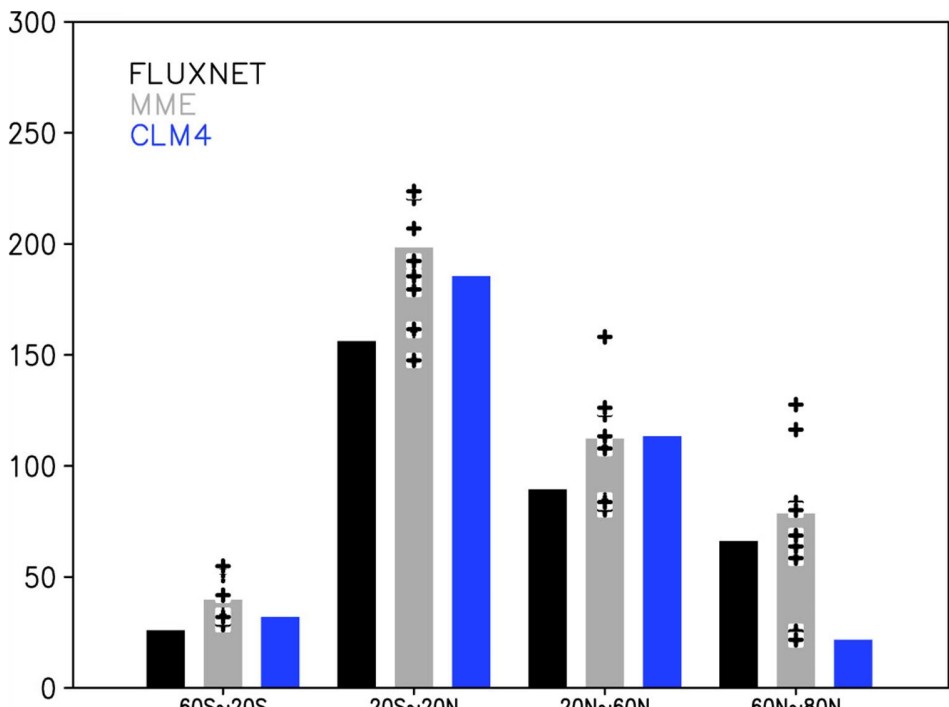

**Figure 4.** Regional averaged GPP in CMIP5 historical runs for 23 years (1983~2005). Black

bars indicate the FLUXNET. Grey bars are MME and symbol dots are individual models. Blue

bars show ESMs which are coupled with CLM4.




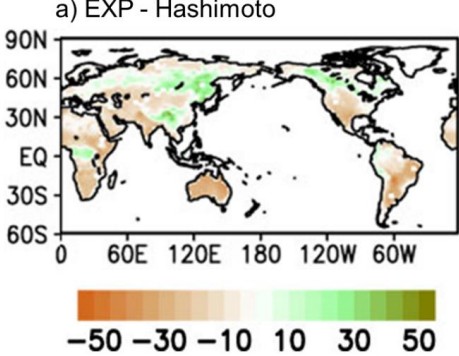

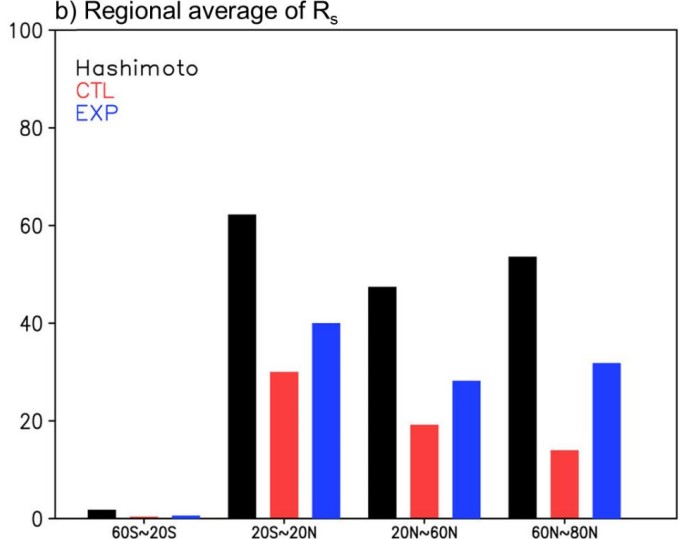


**Figure 5.** (a) shows the spatial distribution of bias pattern of Rs in EXP simulation. (b)

indicates the comparison of the regional average of Rs between Hashimoto data (black bars),

CTL simulation (red bars) and EXP experiment (blue bars).



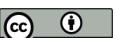



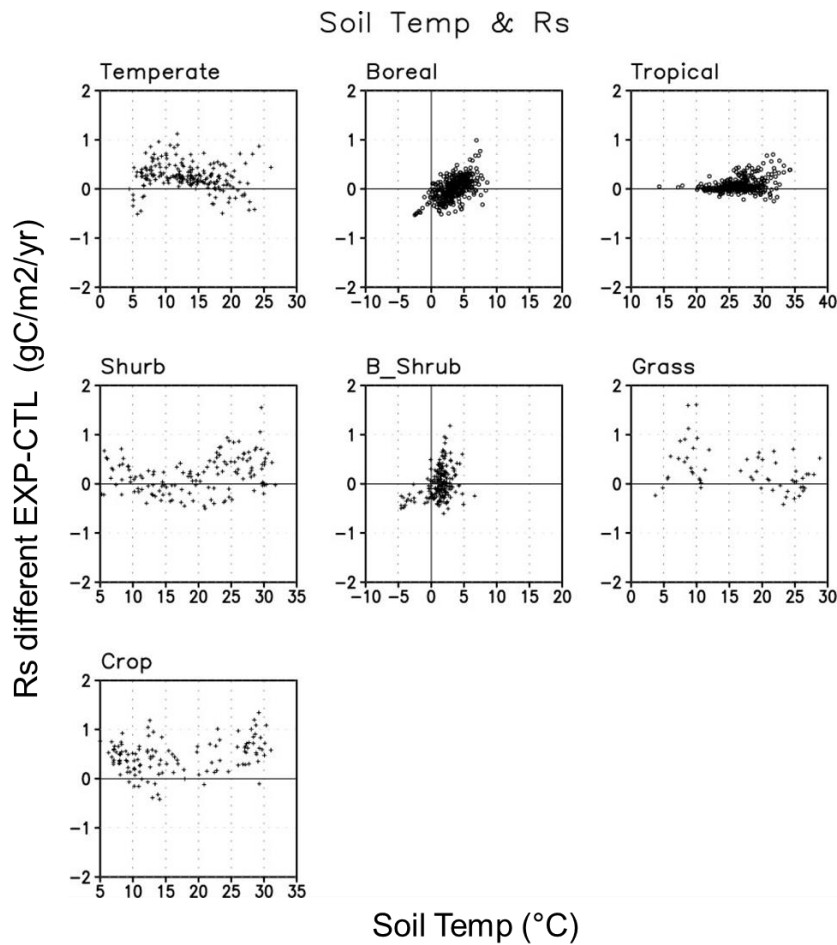


**Figure 6.** Scatter plots of change of Rs (y-axis) between EXP and CTL simulation as a

function of soil temperature (x-axis). Each panel shows the plots for different PFTs that include

temperate (temperate NET and BET), boreal (boreal NET, NDT, BDT), tropical (toprical BET,

BDT), Shrub, B_shrub (Boreal shrub), Grass(Grass) and Crop(Crop).

628



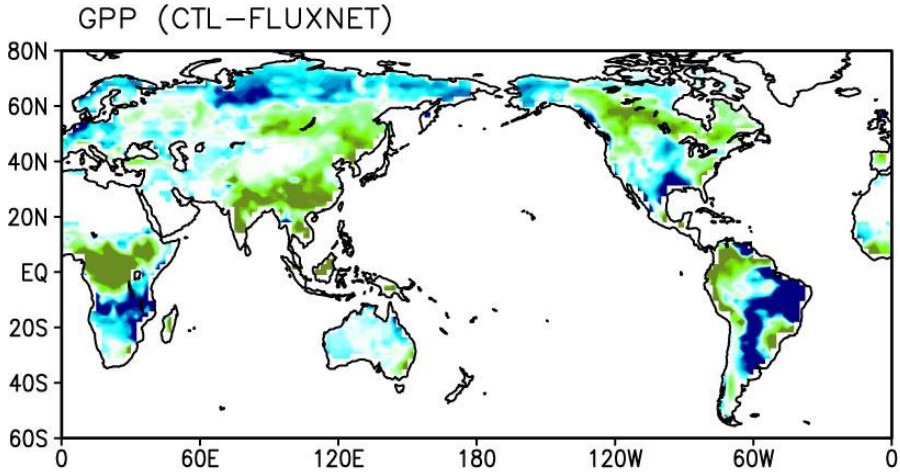

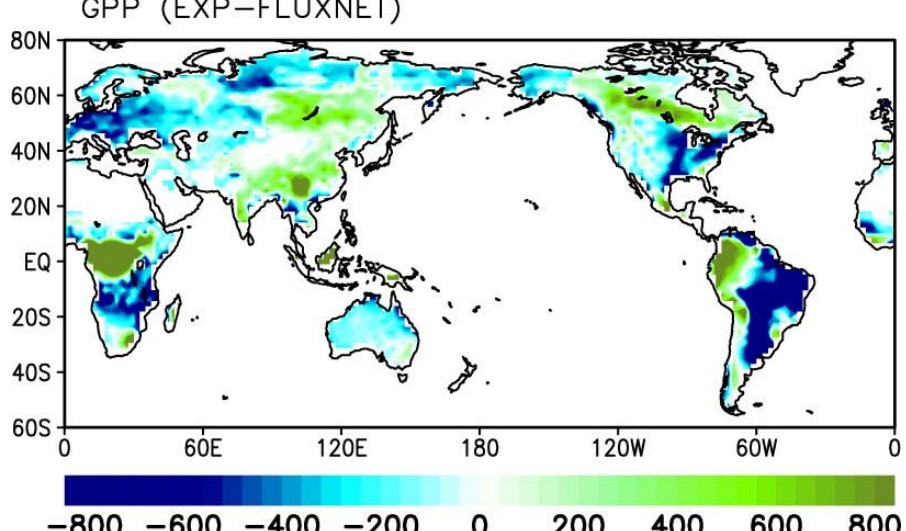

**Figure 7.** Bias of GPP spatial distribution in CTL and EXP comparing with FLUXNET

during 23 years (1983-2005)




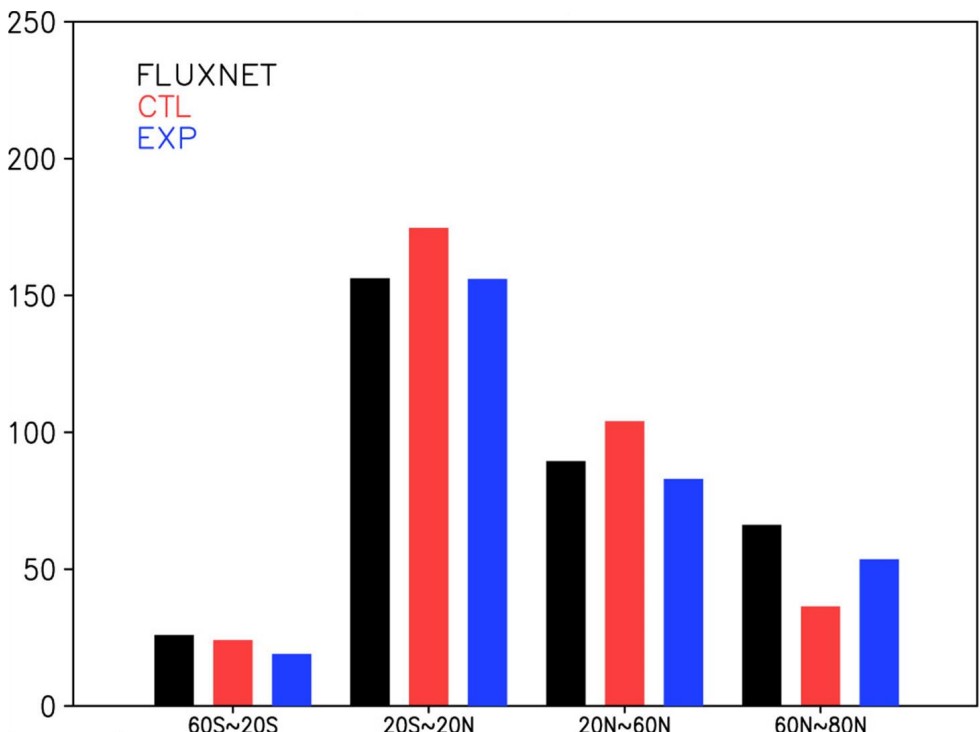


**Figure 8.** Regional averaged GPP in FLUXNET (black bars), CTL (red bars) and EXP (blue
bars).




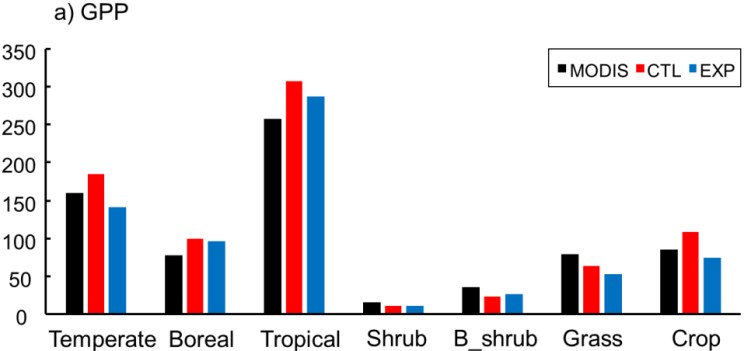

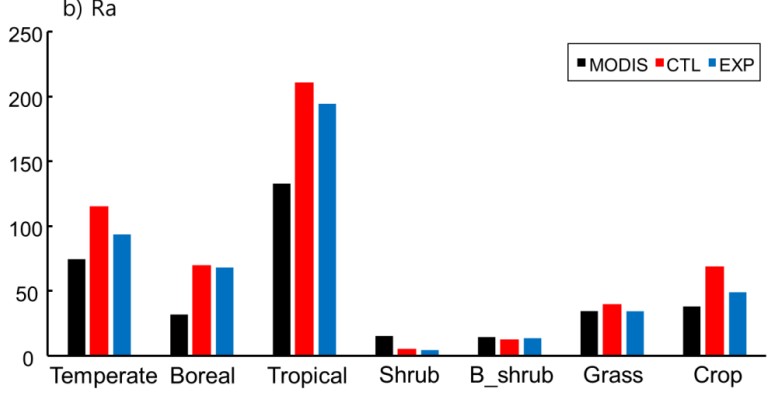

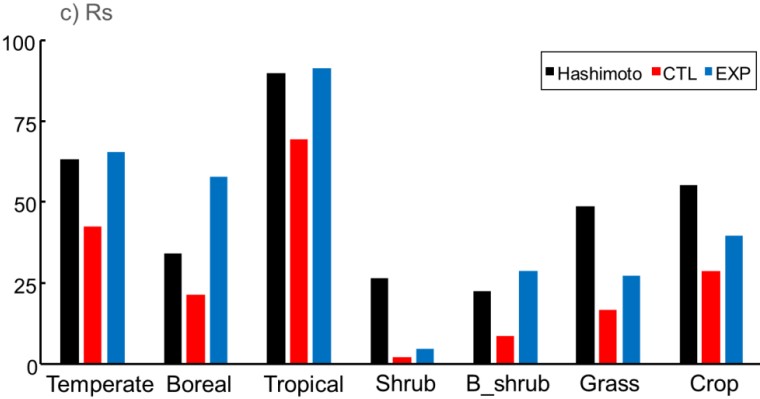


**Figure 9.** Comparison of spatial average of GPP, Ra and Rs in observation (black bars),
CTL (red bars) and EXP (blue bars) by PFTs.