# Peer review of "Published: 12 January 2017"

_Biogeosciences, 2016_

## Referee Comment (RC1) · Anonymous Referee #1 · 9 Feb 2017

Land carbon models are critical for understanding controllers of atmospheric carbon dioxide under a changing climate. As such, accurately estimating soil respiration sensitivity to temperature and moisture is critical. This manuscript presents a re-analysis of existing data products to propose new biome specific parameterizations focusing on temperature effects. Unfortunately I find the manuscript confusing on several points and their main conclusions flawed.

This manuscript uses global data products to examine the temperature and moisture of soil heterotrophic(?) respiration. Given that the authors did not provide their analysis

scripts, and how they presented the variables used in this study, I'm forced to conclude that the soil respiration they used to drive this analysis was, itself, a model (Hashimoto et al., 2015). This makes this study a reanalysis of an existing land carbon model. While that could be interesting, since the parameterization of the Hashimoto Rs data product environmental sensitivity was global and this study proposes biome specific sensitivities of different forms, this does not support the main claims of the study to develop new parameterization. Instead it makes the case that a biome specific model can accurately describe a globally parameterized model.

Other points:

The authors need to clarify how their GPP analysis ties into their main points about soil respiration (which is unclear whether they are referring to root + heterotrophic respiration or solely heterotrophic respiration).

There is considerable controversy in the field over whether Q10 is a global parameter (Karhu et al., 2014; Mahecha et al., 2010), spatially heterogeneous (as cited by the authors) or chemically heterogeneous . The authors need to review this in the introduction, another good reference for the introduction may be (Davidson et al., 2006; Davidson and Janssens, 2006). While I have no problem with a study to examine the implications of a spatially explicit Q10 sensitivity, to frame this as a broad community consensus is incorrect.

The authors lost me on Eq 4 (though the subsequent Eq 4 to 8 progression is well presented). What is q in Eq 4 and how does it relate to the traditional presentation of: Rs = k*C*f(T)*g(M)? This is critical to the study and needs to be painfully clear. How is the current approach different from fitting log(Rs) = log(k)+log(C) + log(f(T)) + log(g(M)) which is what I expected when I hear a linear regression estimate of temperature and moisture sensitivities. While linear regressions are common in the field I'm not clear on what exactly was being regressed where. Code would help in addition to more details on the exact form of the regression in the methods section.

[Figure]

Please make the code available for this study. While it is not appropriate to reproduce the already available public datasets, it is best practice to make the analysis scripts and software available to increase reproducibility. This will also address the question of the exact structure of the regression model used in this study.

Line by line comments:

Conventionally Q10 is written Q_{10} (with subscripts) the authors may wish to consider reformatting to match convention.

Abstract: Is this a paper about Q10, Rs, or GPP? It's ok to consider all of them but that's not how the paper is initially sold in the title and beginning of the abstract. Right now it reads as three separate ideas and very choppy. Consider integrating the abstract by linking the two in the early sentences and then going onto the detailed results for each and then linking them up again in the conclusion.

P3 L83-85 Most ESMs are decoupled, driven by CO2 concentrations instead of a full feedback carbon cycle. Thus the variation in traditional climate parameters (surface temperature and precipitation) is not due to carbon cycle representations as is implied in these lines. Variations in emissions targets are backed out post-hoc generally via carbon budgeting from the associated carbon cycle and CO2 concentration scenario. Thus it's the emissions targets that tend to reflect the land carbon cycle uncertainty not the overall climate. Please make this clear in the paragraph or specify that you are restricting your discussion to emissions driven ESMs (which will give you a slightly different set of references you need to cite).

P3 L90-92 Make it clear you are talking about direct field characterization of global budgets for soil heterotrophic as opposed to indirect carbon budgeting estimates. Right now it reads like no one has ever looked at measuring soil respiration at all which is completely false as the authors go into detail later on.

P5 L119-121 You need a citation to back up this statement. I suggest (Todd-Brown et

al., 2013) for a review of CMIP5 soil carbon models or directly citing the CMIP5 ESM manuscripts themselves.

P6 L157 Please make it clear if you are using the global soil map or underlying data set from (Hashimoto et al., 2015). Given the reference to regridding I'm assuming this is the soil map product (if this is not the case please clarify and disregard the following comments). This is a fatal flaw in this study. While the model used to generate this data product is not explicitly a Q10 model it is also clearly not in situ observations which makes this study a reanalysis of an existing model not a new interpretation of observations as the authors have framed this manuscript.

P11 Sect 3.1 Why are we looking at GPP here? (Anav et al., 2013) Already looked at GPP in the context of FLUXNET, how is this different? This section still seems disconnected from the rest of the results as was mentioned in above comments on the abstract.

P12 L293-296 This seems to belong in the GPP section. Unless you are also applying the Q10 sensitivity analysis to the GPP product in which case you need to be clearer how that ties into the methods section Eq 4-8 soil referred to Rs.

P12 Please avoid the use of acronyms where possible. CTL and EXT break the flow of the manuscript.

P20 L490 Malformated citation (bad first author name)

References: Anav, A., Friedlingstein, P., Kidston, M., Bopp, L., Ciais, P., Cox, P., Jones, C., Jung, M., Myneni, R. and Zhu, Z.: Evaluating the land and ocean components of the global carbon cycle in the CMIP5 Earth system models, J. Clim., 26, 6801–6843, doi:10.1175/JCLI-D-12-00417.1, 2013. Davidson, E. A. and Janssens, I. A.: Temperature sensitivity of soil carbon decomposition and feedbacks to climate change., Nature, 440, 165–173, doi:10.1038/nature04514, 2006. Davidson, E. A., Janssens, I. A. and Luo, Y.: On the variability of respiration in terrestrial ecosystems: moving beyond Q10,

Glob. Chang. Biol., 12(2), 154–164, doi:10.1111/j.1365-2486.2005.01065.x, 2006. Hashimoto, S., Carvalhais, N., Ito, A., Migliavacca, M., Nishina, K. and Reichstein, M.: Global spatiotemporal distribution of soil respiration modeled using a global database, Biogeosciences, 12(13), 4121–4132, doi:10.5194/bg-12-4121-2015, 2015. Karhu, K., Auffret, M. D., Dungait, J. A. J., Hopkins, D. W., Prosser, J. I., Singh, B. K., Subke, J.-A., Wookey, P. A., Ågren, G. I., Sebastià, M.-T., Gouriveau, F., Bergkvist, G., Meir, P., Nottingham, A. T., Salinas, N. and Hartley, I. P.: Temperature sensitivity of soil respiration rates enhanced by microbial community response, Nature, 513(7516), 81–84, doi:10.1038/nature13604, 2014. Mahecha, M. D., Reichstein, M., Carvalhais, N., Lasslop, G., Lange, H., Seneviratne, S. I., Vargas, R., Ammann, C., Arain, M. A., Cescatti, A., Janssens, I. A., Migliavacca, M., Montagnani, L. and Richardson, A. D.: Global convergence in the temperature sensitivity of respiration at ecosystem level, Science (80-. )., 329(5993), 838–840, doi:10.1126/science.1189587, 2010. Todd-Brown, K. E. O., Randerson, J. T., Post, W. M., Hoffman, F. M., Tarnocai, C., Schuur, E. A. G. and Allison, S. D.: Causes of variation in soil carbon simulations from CMIP5 Earth system models and comparison with observations, Biogeosciences, 10(3), 1717–1736, doi:10.5194/bg-10-1717-2013, 2013.

---

## Referee Comment (RC2) · Anonymous Referee #2 · 12 Mar 2017

Q10 is a critical parameter for simulating soil respiration and C cycle and therefore feedback between C cycle and climate. Yet most ESMs adopt constant Q10, which can possibly lead to unrealistic simulations of soil processes and other C cycle processes. Therefore, this paper contributes to improve model performance by implementing a new Q10 parametrization. This has significant implications to modeling studies.

I have two major concerns. One is the authors did not explain why improvement of Q10 parameterization and resulting improvement of soil respiration can help improve simulations in GPP. Is it because nitrogen availability resulting from changed soil respiration

rates or other mechanisms? Another is the confusion of sensitivity of soil respiration to temperature (i.e., Q10) and sensitivity of Q10 to parametrization.

Specific comments are listed below.

Lines 84-85. Please cite literature(s) for this statement.

Lines 86-87. Please cite literature(s) for this statement too.

Lines 89-90. There are many studies on soil respiration under experimental warming that which should address this point better.

Lines 93-94. There is a recent review paper talking about this issue (Global Biogeochemical Cycles, 2016, 30: 40-56)

Lines 102-103. I would not say that because there are many field studies examined Q10, but it is true, we lack long-term observation data on how Q10 changes overtime. It may be difficult for regular field studies to explore dependency of Q10 on temperature as they derive Q10 through entire seasonal temperature range when Rs is measured. However, it can be tested via lab incubation experiments or field experiments with warming treatments.

Lines 104-105. I think they meant Rs tends to decrease with temperature increase. This needs to be clearly stated.

Line 135. Both the name of the ESM and the name of land model in the ESM should be given.

Line 145. Add "time resolution/interval/step" before "is monthly".

Line 151. "Each tile is 1200 X1200 km" needs to be clearer. Does it refer to original MODIS17A3 GPP and NPP data?

Lines 155-156. Which year are these data for or are they the average from 2000 to 2006?

Line 157. This sentence is not clear. Did they mean that they compared modeled Rs against the data by Hashimoto et al. (2015) for validating their model? What is time period of Rs data by Hashimoto et al. (2015)?

Line 159. What is "SRDB"? It needs to be fully spelt.

Line 162. Year for the literature should be given.

Lines 165-167. What does "assuming" mean?

Lines 176-177. Why is the decomposition flux calculated by multiplying carbon amount from dead leaf? Soil respiration include both microbial respiration and root respiration. The substrate for microbial respiration is SOM in the soil, which is originally derived from litter (dead leaf, wood, and root). And root respiration is respiration by live roots, which is related to root biomass. If their model does not separate the two components, it should be carbon content of soil.

Lines 186-187. It should be "the water potential for soil decomposition".

Lines 212-213. In a multiple regression analysis, why are the relationships between Rs and T and between Rs and M separately?

Line 223. Use soil temperature may be clearer than subsurface data.

Lines 224-235. This paragraph is very confusing. The authors need to give time step of each dataset to avoid confusion. Are GPCP and TRMM data used to rescale precipitation data by Sheffield et al. (2006) to a time step of daily and 3-hour step for forcing CLM? And CLM is forced by 3hr data, then why daily data are needed? Is it for regression analysis? In addition, Sheffield et al. (2006) data have radiation, so what are radiation data by NASA for? If they are for all descriptions of Sheffield et al. (2006), it needs to be clear.

Line 226. Year for the literature is needed.

Lines 233-235. Please give the source of the data.

Lines 235-237. What does "integration" mean? I understand it is model simulation.

Lines 236-237. How can an audience find a reference that is not published?

Line 238. I think they are talking about soil respiration by Hashimoto et al. (2015), but this needs to be specified.

Lines 246-247. The period of forcing data by Sheffield et al. (2006) is reduced to be the same with GSWP2?

Line 251. Does this mean a global constant and unchanged over time? Fig. 1. Why 28 years? Is this because of time period of data by Hashimoto et al. (2015)? The title is confusing and needs to be revised. It would be useful for audiences to see regression models for all PFT as supplementary information.

Line 252. CESM run needs to be described before "Figure 2".

Fig. 3. CTL should be included in the figure title.

Line 272. The title should be "results and discussions".

Line 330. Sensitivity of Rs to soil temperature?

Line 334. Which panel in Fig. 6 did they refer to by "enhanced relationship between Rs and temperature" for the northern Eurasian and Chinese regions?

Line 273 and the whole paragraph. In method section they never talked about CMIP5, why here a subtitle for CMIP5 GPP? If they use CMIP5 to evaluate or compare CLM EXP, they needs to describe it in method and need to include EXP or CTL in this section and in Fig. 4. They also need to give some details such as how many CMIP5 models, model names and if MME includes CESM-BGC and NorESM. There are few papers with figures that do not include results from their present study. If they want to discuss the issue of underestimation n GPP by coupled N cycle, this part should be put into discussion and the figure should be in the supplementary document. Overall, this part is not very relevant to the main purpose of this study.

Figure 4. Figure title needs more information. "MME" needs to be fully spelt. I would use "CMIP5" instead of "MME" in the legend. Are blue bars the average of CESM-BGC and NorESM? Why no results from EXP? Global GPP can also be shown in this figure since it is mentioned in the text. In addition, no y axis (unit) in this figure.

Line 288. What are they talking about by "these two"? In addition, according to the figure, GPP 60N-80N is not major region.

Line 293. This should be Rs, not GPP.

Lines 293-296. Delete this since it is a repeat of last paragraph.

Lines 296-300. How could they conclude that from Fig. 5 since a) is difference between EXP and observation, not the absolute values of observation and EXP. They can show maps of all three data sets (observation, EXP and CTL) in supplementary documents to support this statement.

Fig. 5. I would suggest to add another panel for the difference between CTL and observation. Unit is missing for both panels.

Lines 304-305. I did not see this in Fig. 5.

Lines 293-331. It would be easier to give panel number such as "Fig 5a)".

Lines 310-311 Increase compared to CTL and underestimation compared to observation should be stated clearly.

Lines 312-313. Fig. 6 does not support this point because the y axis is the difference between EXP and CTL. It is not the absolute Rs of EXP or CTL. More changes do not necessarily result in higher absolute Rs. The point may be supported if they draw scatter plots for both EXP and CTL in each panel and show better correlation between Rs and temperature in EXP than CTL.

Line 314. "The difference between EXP and CTL increases with temperature" is not supported by Fig. 6 since they are only the cases in a few panels.

Fig. 6. The categories are confusing. Some are regions such as temperate but some are ecosystem types such as grass. P values are needed for the correlations. The abbreviations should be fully spelt.

Lines 317-318. This reason is not convincing.

Lines 319-322. This explanation is not convincing because tropical is the opposite and it cannot explain shrub, grass and crop.

Fig. 7. Unit should be given.

Lines 336-334. Please explain the mechanism for this.

Lines 337-340. The global GPP in FLUXNET should also be given here. "SH" should be fully spelt.

Line 344. Are the numbers in Fig. 8 zonal mean or zonal sum? I think they are sums.

Line 345. The word "budget" is not suitable here. Use GPP.

Line 348. Are they talking about Fig. 3? Please indicate.

Fig. 8. Adding global data to this figure would help. This figure should merged with Fig. 7 (i.e., three panels). Y axis is missing.

Lines 349-350.What did the authors mean here? How can carbon pool in the soil system affect plant assimilation? Plants do not absorb carbon in soil.

Fig. 9. No y axis.

Lines 393-394. This has never been mentioned in the text. Why are they interested in this ratio? I do not see any implications of this ratio.

Lines 398-400. Please cite literature here.

Lines 404-405. This sentence is not clear.

---

## Referee Comment (RC3) · Anonymous Referee #3 · 20 Mar 2017

The authors developed PFT-dependent Q10 values for soil organic matter (SOM) decomposition processes using a multiple regression method. They demonstrated that the spatially-distributed Q10 had the potential to improve the simulation of both soil respiration and GPP compared with the CLM4 simulation with a uniform Q10. It's necessary and important to use spatially-distributed Q10 rather than a constant Q10 in global simulations. I would like the authors to further clarify the "multiple regression" method used in this study as I don't quite understand it while reading the manuscript: (1) what are the response variables (Rs?) and explanatory variables (T & M?) in the regression analysis? (2) what datasets at what time-scale are used for regression?

(3) what is the relation between the equations 4-8 and the regression analysis? (4) how do you calculate Q10 at every time interval as you stated in Line 381? Q10 is temperature-dependent as indicated in Eqs. 2 & 5, do you mean that you will also change Q10 based on the temperature at current time-step? Another concern of mine is related to the calculation of Q10 using soil respiration data. We know that generally soil respiration includes both heterotrophic respiration from SOM decomposition and root respiration (growth + maintenance). It seems the PFT-dependent Q10 is developed for SOM decomposition processes, thus how do you use total soil reparation to determine the Q10 for SOM decomposition?

Minor comments: (1) Fig.5 & Fig. 9: please indicate the units of Rs and Ra. In addition, please explain what are Ra and Rs, i.e., plant autotrophic respiration and soil respiration. (2) Figs.4, 7, 8 & 9: please indicate the units of GPP. (3) Line 304: "The Rs Simulation difference between CTL and EXP is given in Figure 5, in terms of global distribution as well as zonally-averaged distribution". I understand we may identify the zonal difference between CTL and EXP. However, Fig.5a shows the difference between EXP and Hashimoto data, not between EXP and CTL. (4) Line 314: "the difference between EXP and CTL increases with temperature". It may be true for boreal and B_Shrub PFTs. I would suggest doing statistical tests to show whether the relation is significant or not.

---

## Referee Comment (RC4) · Anonymous Referee #4 · 21 Mar 2017

I fear I cannot really write a more positive review for this very manuscript. I have to say that I am quite confused by this paper. In particular, I have two very fundamental concerns:

1. The authors write in the abstract "... (non-uniform spatial distribution of $Q_{10}$) ... improves the simulation of gross primary production (GPP). It leads to a more realistic spatial distribution of GPP, particularly over high latitudes ...". This statement suggests that GPP$= f(Q_{10,soil}, ...)$ which makes no sense at all to me. Or do I fundamentally misunderstand the model assumptions here? To me this sen-

tence suggests that either I don't understand the basic dynamics under scrutiny, or that the authors have been very sloppy in putting the manuscript together, or that there is indeed a very fundamental conceptual issue here. I fear that we are talking about the latter.

2. The authors write that "the $Q_{10}$ value derived from soil respiration measurement tends to decrease with temperature because substrate availability decreases as temperature increases". To my mind this is rather reflecting that any regression model that considers abiotic drivers only, would be confounded by co-variations with e.g. substrate supply or other biotic drivers. This has been shown e.g. in Reichstein & Beer (2008), Mahecha et al. (2010), Wang et al. (2010), Graf et al. (2011), and we could cite more recent papers. Inferring from varying abiotic controls that $Q_{10}$ should also vary across geographic locations is misleading. Non-constant parameters in biosphere models may actually reflect missing process detail: For instance, if one tries to subsume in $Q_{10}$ variation of e.g. substrate supply then one is simply missing a good representation of the supply term. Various papers by e.g. Davidson (e.g. 2012 and more recent ones) explain this in a very didactic manner and should be studied before discussing this aspect further and tweaking models without actually developing the underlying model structures further.

To my mind, the authors first need to clarify these two aspects in a very convincing manner before discussing the "minor" issues of the manuscript. However, these other aspects that are in fact not so minor. For instance when the authors write that "the $Q_{10}$ parametrization tends to enhance the relationship between Rs and soil temperature from CTL" they refer to Fig. 6 which show differences in respiration modelled with different runs and $T_{soil}$. But the scatters are all over the place (positive/negative) and it is unclear about what relationship we are talking here. And I find more examples of this kind in the text . . ..

So in the overall view, I would like to encourage the authors to carefully rethink what the focus of this study can be and what can be really learned with this experiments.

**References** Davidson, E. A. et al. (2012) *The Dual Arrhenius and Michaelis–Menten kinetics model for decomposition of soil organic matter at hourly to seasonal time scales.* Global Change Biology, 18, 371–384.

Graf, A. et al. (2011) *Comment on "Global convergence in the temperature sensitivity of respiration at ecosystem level".* Science, 331, 1265.

Mahecha, M. D. et al. (2010) *Global convergence in the temperature sensitivity of respiration at ecosystem level.* Science, 329, 838–840.

Reichstein, M. and Beer, C. (2008) *Soil respiration across scales: The importance of a model-data integration framework for data interpretation.* Journal of Plant Nutrition and Soil Science, 171, 344–354.

Wang, X. et al. (2008) *Are ecological gradients in seasonal $Q_{10}$ of soil respiration explained by climate or by vegetation seasonality?.* Soil Biology & Biochemistry, 42, 1728–1734.

---

## Author Comment (AC1) · 7 May 2017

I fear I cannot really write a more positive review for this very manuscript. I have to say that I am quite confused by this paper. In particular, I have two very fundamental concerns:

⇒ We appreciate the reviewer's critical comments. We carefully address to the points made by the reviewer and try to clarify them in the below.

[Figure]

1. The authors write in the abstract "... (non-uniform spatial distribution of Q10) ... improves the simulation of gross primary production (GPP). It leads to a more realistic spatial distribution of GPP, particularly over high latitudes ... ". This statement suggests that GPP= f(Q10;soil; ...) which makes no sense at all to me. Or do I fundamentally misunderstand the model assumptions here? To me this sentence suggests that either I don't understand the basic dynamics under scrutiny, or that the authors have been very sloppy in putting the manuscript together, or that there is indeed a very fundamental conceptual issue here. I fear that we are talking about the latter.

⇒ Soil respiration (Rs) is one of the critical processes for maintaining a terrestrial ecosystem (L49-50), and also important in closing global carbon cycle (L83-85). The process also affects GPP indirectly through the soil decomposition flux of carbon at the root zone (L174-175) that affect plant assimilation. ⇒ The CLM4 model tested in this study uses the carbon-nitrogen (C-N) coupling in the parameterization of terrestrial carbon fluxes. Unlike the other ESMs, the interactive C-N cycle implemented in CLM4 acts as a limiting factor for photosynthesis and gross primary production (GPP) in this model.

⇒ The modification of Q10 in CLM4 tends to modify the carbon decomposition flux in the soil layers. The non-uniform spatial distribution of Q10 directly changes the decomposition flux in the soil (Eq. (1) & (2) in the manuscript) and heterotrophic respiration by root and soil organic matters. The parameterized process is such that a higher Q10 produces faster carbon decomposition. This increased carbon decomposition tends to increase nitrogen flux into the soil from debris, and hence increase the nitrogen assimilation from soil to vegetation. These processes tend to improve the simulation of GPP distribution by modified Q10 parameterization of CLM4.

2. The authors write that "the Q10 value derived from soil respiration measurement tends to decrease with temperature because substrate availability decreases as temperature increases". To my mind this is rather reflecting that any regression model that considers abiotic drivers only, would be confounded by co-variations with e.g. substrate

supply or other biotic drivers. This has been shown e.g. in Reichstein & Beer (2008), Mahecha et al. (2010), Wang et al. (2010), Graf et al. (2011), and we could cite more recent papers. Inferring from varying abiotic controls that Q10 should also vary across geographic locations is misleading. Non-constant parameters in biosphere models may actually reflect missing process detail: For instance, if one tries to subsume in Q10 variation of e.g. substrate supply then one is simply missing a good representation of the supply term. Various papers by e.g. Davidson (e.g. 2012 and more recent ones) explain this in a very didactic manner and should be studied before discussing this aspect further and tweaking models without actually, developing the underlying model structures further.

⇒ First, whether Q10 is a global constant or variable across the abiotic conditions as well as the type of vegetation is controversial. We admit it and will insert this issue in the revised manuscript as below: (From L102∼) "Whether this value is a global constant or variable in space is still under debate and the conclusions from the previous studies are diverse, which reflect our limited understanding to the soil respiration process. For example, Mahecha et al. (2010) suggested that the Q10 value is independent of mean annual temperature and biomes. Karhu et al. (2014) also mentioned that the Q10 is approximately a global constant about 1.4 in the high latitude regions in the northern hemisphere. Another studies, on the other hand, suggested that Q10 may vary in space (Zhou et al., 2009; Xu and Qi, 2001; Qi et al., 2002)."

⇒ Secondly, we admit other biotic and abiotic factors may complicate the process and result in spatial variability instead of Q10, as the reviewer criticizes. But, from the parameterization point of view, the concept of Q10 is implemented differently across the state-of-the-art global prediction models, rather indirectly or implicitly in representing the dependence on biotic and abiotic conditions.

⇒ Like the formula in Eq. (2) & (6), the Q10 value in the CLM4 land surface model used in this study has a much simpler form and inclusive in counting on abiotic conditions such as soil temperature and moisture. These dependences can be also plant functional type (PFT) dependent. While the original version of CLM4 uses a constant value of 1.4, this study attempts to use variable Q10 values depending on PFT and abiotic conditions. Due to its much simpler form of parameterizations, those dependences are not to be represented in other means.

⇒ To be a little more detail, the soil decomposition of the plant detritus in CLM4 depends on the climate conditions as well as on substrate nutrients such as carbon and nitrogen ratios and their amounts. CLM4 considers biotic drivers in the carbon decomposition processes indirectly. These soil carbon dynamics are based on the kinetic theory of biological reaction which are related with soil temperature and moisture. As the reviewer commented, Davidson and Janssens (2006) and other studies used more comprehensive formulations with more biotic and abiotic drivers such as soil aggregation and chemical protection of soil organic matter in addition to temperature and moisture. Biotic drivers are not considered at all or too much simplified in the existing ESMs like CLM4.

⇒ This approach seems to be feasible particularly for the use of climate change experiments, in which the temperature sensitivity may not be static in a warmer climate. It is also arguable to use a constant Q10 value for a specific biome, having said that there may exist a substantial variation of subsurface temperature and soil moisture even within a same type of PFT, either zonally or meridionally. This is often the case of most ESMs where the biomes are represented by several dominant PFTs in a global domain due to a coarse spatial resolution. These were the main motivations for the new parameterization of Q10. To my mind, the authors first need to clarify these two aspects in a very convincing manner before discussing the "minor" issues of the manuscript. However, these other aspects that are in fact not so minor. For instance when the authors write that "the Q10 parametrization tends to enhance the relationship between Rs and soil temperature from CTL" they refer to Fig. 6 which show differences in respiration modelled with different runs and Tsoil. But the scatters are all over the place (positive/negative) and it is unclear about what relationship we are talking here. And I

find more examples of this kind in the text . . .

⇒ Figure 6 shows the scatter plots for Rs change (EXP minus CTL in y-axis), as a function of soil temperature (x-axis). Presumably due to the sub-biome variability in biotic and abiotic conditions, the scatter plots exhibit some nonlinearity in the curvature, but most of the values lie in the positive range for the moderate to warm temperatures (i.e., increased Rs at the given temperature by the variable $Q10$ formulatoin). Note that this relationship is not uniform in space, as the change of $Q10$ is not uniform in EXP (as shown in Fig. 2).

So in the overall view, I would like to encourage the authors to carefully rethink what the focus of this study can be and what can be really learned with this experiments.

⇒ We appreciated your crucial comments for overcoming scientific deficiencies in the manuscripts.

Davidson, E. A. and Janssens, I. A.: Temperature sensitivity of soil carbon decomposition and feedbacks to climate change., Nature, 440, 165–173, doi:10.1038/nature04514, 2006. Davidson, E. A., Janssens, I. A. and Luo, Y.: On the variability of respiration in terrestrial ecosystems: moving beyond Q10, Glob. Chang. Biol., 12(2), 154–164, doi:10.1111/j.1365-2486.2005.01065.x, 2006. Karhu, K., Auffret, M. D., Dungait, J. A. J., Hopkins, D. W., Prosser, J. I., Singh, B. K., Subke, J.-A., Wookey, P. A., Ågren, G. I., Sebastià, M.-T., Gouriveau, F., Bergkvist, G., Meir,P., Nottingham, A. T., Salinas, N. and Hartley, I. P.: Temperature sensitivity of soil respiration rates enhanced by microbial community response, Nature, 513(7516), 81–84, doi:10.1038/nature13604, 2014. Mahecha, M. D., Reichstein, M., Carvalhais, N., Lasslop, G., Lange, H., Seneviratne, S. I., Vargas, R., Ammann, C., Arain, M. A., Cescatti, A., Janssens, I. A., Migliavacca, M., Montagnani, L. and Richardson, A. D.: Global convergence in the temperature sensitivity of respiration at ecosystem level, Science (80-. )., 329(5993), 838–840, doi:10.1126/science.1189587, 2010. Qi, Y., Xu, M., and Wu, J.: Temperature sensitivity of soil respiration and its effects on ecosystem

carbon budget: nonlinearity begets surprises, Ecolog. Model., 153, 131–142, 2002 Xu, M., and Qi, Y.: Spatial and seasonal variations of Q10 determined by soil respiration measures at a Sierra Nevadan forest, Global Biogeochem. Cy., 15, 687 – 696, 2001. Zhou, T., Shi, P., Hui, D., and Luo, Y.: Global pattern of temperature sensitivity of soil heterotrophic respiration (Q10) and its implications for carbon-climate feedback, J. Geophys. Res., 114

Please also note the supplement to this comment:
http://www.biogeosciences-discuss.net/bg-2016-549/bg-2016-549-AC1-supplement.zip
* * *

---

## Author Comment (AC2) · 7 May 2017

Land carbon models are critical for understanding controllers of atmospheric carbon dioxide under a changing climate. As such, accurately estimating soil respiration sensitivity to temperature and moisture is critical. This manuscript presents a re-analysis of existing data products to propose new biome specific parameterizations focusing on temperature effects. Unfortunately I find the manuscript confusing on several points and their main conclusions flawed.

We appreciate the reviewer's valuable comments. We carefully address to the points made by the reviewer and try to clarify them in the below.

This manuscript uses global data products to examine the temperature and moisture of soil heterotrophic(?) respiration. Given that the authors did not provide their analysis scripts, and how they presented the variables used in this study, I'm forced to conclude that the soil respiration they used to drive this analysis was, itself, a model (Hashimoto et al., 2015). This makes this study a reanalysis of an existing land carbon model. While that could be interesting, since the parameterization of the Hashimoto Rs data product environmental sensitivity was global and this study proposes biome specific sensitivities of different forms, this does not support the main claims of the study to develop new parameterization. Instead it makes the case that a biome specific model can accurately describe a globally parameterized model.

Soil respiration (Rs) is one of the critical processes for maintaining a terrestrial ecosystem (L49-50), and also important in closing global carbon cycle (L83-85). The process also affects GPP indirectly through the soil decomposition flux of carbon at the root zone (L174-175) that affect plant assimilation. The conventional ESMs count on the sensitivity of Rs to the soil temperature using a constant value of Q10, but it is globally uniform, regardless of plant function types (PFTs).

The novelty of this study is to develop a new parameterization method for Q10, instead of using a fixed value in the conventional ESMs. There may exist different concepts for "parameterization". To be faithful to the concept of "parameterization" in the earth system modeling, the new parameterization in this study determines the Q10 value dynamically depending on abiotic conditions (e.g., subsurface soil temperature and moisture) as well as depending on PFTs. This approach seems to be feasible particularly for the use of climate change experiments, in which the temperature sensitivity may not be static in a warmer climate. It is also arguable to use a constant Q10 value for a specific biome, having said that there may exist a substantial variation of subsurface temperature and soil moisture even within a same type of PFT, either zonally or

meridionally. This is often the case of most ESMs where the biomes are represented by several dominant PFTs in a global domain due to a coarse spatial resolution. These were the main motivations for the new parameterization of Q10.

One may use either theoretical or empirical approach to derive the relationship between Rs and the abiotic conditions. The parameterization of Q10 in this study is based on the empirical relationship between Rs and subsurface temperature and moisture for the given PFT. In doing this, the crucial part is the quality of the reference data and the degree of fitting (L220-221). Regarding the data, as the reviewer criticizes, the practical problem here is there is no real observation data for soil respiration and subsurface temperature and moisture. Although in-situ data of soil respiration are available from the Soil Respiration Database (SRDB, Bond-Lamberty and Thomson, 2010), the data have limited sampling for boreal cold regions (i.e., tundra and northern Siberia) as well as unpopulated regions in the tropics, covering a significant portion of the global biosphere (L159-162). Same problems lie in the subsurface data for temperature and moisture, as there is no comprehensive observation data covering the globe. Recent satellite instruments using microwave channels can retrieve subsurface soil moisture, but this is all limited in spatial and temporal sampling.

As a reference for observations, this study used the re-analysis soil respiration data from Hashimoto et al. (2015). Although the data were derived using an empirical soil respiration model based on Raich et al. (L162-169), they are not entirely modeled data but using the SRDB observation data. This study also conducted an "independent" reanalysis for the subsurface soil temperature and moisture by integrating the land surface model driven by observational forcing for a sufficiently long period (e.g., 1983-2010), which was to better represent the subsurface climatology at the presence of strong interannual variability.

We admit our approach used another modeled data, but the use of reanalysis is the best alternative choice when the exact in-situ data are not available such as in our case. The current parameterization method can be further improved by calibrating

the empirical relationship between Rs and soil moisture and temperature, once the exact data from observations are available. The new parameterization for Q10 also demonstrates a good degree of fitting. The biome specific sensitivity for each 17 plant functional types (PFTs) presented in Fig. 1 shows a good skill of Q10 parameterization for the simulation of Rs, even though the soil respiration data of Hashimoto et al. and the off-line land surface model data from this study were produced independently.

For clarification, we attach the Excel file showing data and the regression results between Rs and soil temperature and moisture for each 17 PFTs.

Other points: The authors need to clarify how their GPP analysis ties into their main points about soil respiration (which is unclear whether they are referring to root + heterotrophic respiration or solely heterotrophic respiration).

The soil respiration in this study refers to the sum of root and heterotrophic respiration. The Q10 parameterization in the model changes the decomposition rate of carbon by soil organic matter in the soil layers. Based on the Equations (1) and (2), a higher Q10 value tends to increase carbon decomposition (and nitrogen) into soil layers, which tends to enhance the nitrogen assimilation to plants.

The experiment with the variable Q10 parameterization tends to increase GPP in the northern hemisphere high latitudes (Fig. 7 & 8) where Q10 increases (Fig. 2), and decrease GPP in the tropics and midlatitudes where Q10 decreases. This suggests the change in soil respiration affects GPP through the plant assimilation process.

We further examined the impact of Rs on plants net primary production. The variable Q10 parameterization tends to affect the turnover time of soil carbon, which is defined as the soil carbon amount divided by net primary production (NPP) (i.e., soil carbon/NPP). As shown in Fig. S1 below, the run with variable Q10 (EXP) makes shorter turnover time in northern hemisphere high latitudes and longer in the tropics compared with the control run(CTL). Shorter turnover time in high latitudes suggests the enhancement of nitrogen assimilation to vegetation in EXP, thereby enhancing net

primary production by plants.

There is considerable controversy in the field over whether Q10 is a global parameter (Karhu et al., 2014; Mahecha et al., 2010), spatially heterogeneous (as cited by the authors) or chemically heterogeneous . The authors need to review this in the introduction, another good reference for the introduction may be (Davidson et al., 2006; Davidson and Janssens, 2006). While I have no problem with a study to examine the implications of a spatially explicit Q10 sensitivity, to frame this as a broad community consensus is incorrect.

We totally agree with the reviewer, and we will reflect the reviewer's comment in the revised manuscript as below:

(From L102∼) "Whether this value is a global constant or variable in space is still under debate and the conclusions from the previous studies are diverse, which reflect our limited understanding to the soil respiration process. For example, Mahech et al. (2010) suggested that the Q10 value is independent of mean annual temperature and biomes. Karhu et al. (2014) also mentioned that the Q10 is approximately a global constant about 1.4 in the high latitude regions in the northern hemisphere. Another studies, on the other hand, suggested that Q10 may vary in space (Zhou et al., 2009; Xu and Qi, 2001; Qi et al., 2002)."

The authors lost me on Eq 4 (though the subsequent Eq 4 to 8 progression is well presented). What is q in Eq 4 and how does it relate to the traditional presentation of: Rs = k*C*f(T)*g(M)? This is critical to the study and needs to be painfully clear. How is the current approach different from fitting log(Rs) = log(k)+log(C) + log(f(T)) + log(g(M)) which is what I expected when I hear a linear regression estimate of temperature and moisture sensitivities. While linear regressions are common in the field I'm not clear on what exactly was being regressed where.

"q" in Eq. (4) represents the "fractional" change of Rs due to temperature, which can be decomposed further into the sensitivity to the soil moisture and temperature as in Eq.

(8). The reason why we used the fractional change is in that Rs is assumed to be an exponential function of temperature and moisture. This enables one to take logarithm to Rs and expand it to the sum of individual sensitivities, as in the one that the reviewer commented.

Using Eq. (8), we constructed the multiple regression equation for Rs with respect to the changes in T and M. The sensitivity of Rs to temperature has been known to be exponential in the previous studies. The traditional equation of soil respiration is defined by van't Hoff (1898) :

Resp=$\alpha$e$^\beta$T, (a)

where $\alpha$, $\beta$ are fitted parameters for regression. This Van't Hoff equation is modified by Davidson et al. (2006) as:

Resp=ãĂŰR_base Q_10ãĂŮˆ(((T-T_base)/10)), (b)

where T and Tbase are measured temperatures. The subscripts "base" indicates the base state at the specific time. Q10 is defined as the factor of respiration variation by 10 kelvin degree temperature increasing.

Code would help in addition to more details on the exact form of the regression in the methods section. Please make the code available for this study. While it is not appropriate to reproduce the already available public datasets, it is best practice to make the analysis scripts and software available to increase reproducibility. This will also address the question of the exact structure of the regression model used in this study.

We provided the excel file which was used to obtain the regression results between soil respiration and temperature and moisture in each plant functional types (PFTs). Please check attached excel files.

Line by line comments:

Conventionally Q10 is written Q_{10} (with subscripts) the authors may wish to consider reformatting to match convention.

We will modify as Q10 (subscripts) in numerous places in the manuscript. .

Abstract: Is this a paper about Q10, Rs, or GPP? It's ok to consider all of them but that's not how the paper is initially sold in the title and beginning of the abstract. Right now it reads as three separate ideas and very choppy. Consider integrating the abstract by linking the two in the early sentences and then going onto the detailed results for each and then linking them up again in the conclusion.

We will revise the abstract carefully following the reviewer's comment. The first two sentences will be modified as:

(L49-53) "Soil decomposition is one of the critical processes in maintaining terrestrial ecosystem and global carbon cycle. Soil respiration (Rs) sensitivity to temperature so called the Q10 value required for parameterizing soil decomposition process is assumed to be a constant in conventional numerical models, while it is not so in the realistic case with spatiotemporal heterogeneity." Also we revise the introduction part the linkage between Rs and GPP in the manuscript: (L130) "Realistic spatial distribution of soil decomposition processes affect not only Rs but also primary production by improving nitrogen assimilation from soil to vegetation."

In CLM4 (Olsen et al., 2013), plant nitrogen uptake from soil mineral nitrogen pool is separated by plant demand for mineral nitrogen from the soil (NFplant_demand_soil) and retranslocated nitrogen (NFretrans) which construct to mobilize senescing tissues. Therefore, total plant nitrogen uptake from soil mineral nitrogen pool is : ãĂŰN-FãĂŮ_(plant_demand_soil)= ãĂŰNFãĂŮ_(plant_demand)-ãĂŰNFãĂŮ_retrans

This total plant nitrogen demand for new growth (NFplant_demand) is calculated by total carbon available for new vegetation growth allocation (CFavail_alloc) from soil as:

ãĂŰNFãĂŮ_(plant_demand) ãĂŰCFãĂŮ_(avail_alloc) N_allom/C_allom

where CFavail_alloc is related with carbon amount in each carbon pools. These processes induce that more carbon decomposition enhanced more nitrogen supplement from soil to plant for new plant growth (increasing GPP).

P3 L83-85 Most ESMs are decoupled, driven by CO2 concentrations instead of a full feedback carbon cycle. Thus the variation in traditional climate parameters (surface temperature and precipitation) is not due to carbon cycle representations as is implied in these lines. Variations in emissions targets are backed out post-hoc generally via carbon budgeting from the associated carbon cycle and CO2 concentration scenario. Thus it's the emissions targets that tend to reflect the land carbon cycle uncertainty not the overall climate. Please make this clear in the paragraph or specify that you are restricting your discussion to emissions driven ESMs (which will give you a slightly different set of references you need to cite).

Following the reviewer's comment, we will modify the sentences as: (L83-84) : "Future climate change projection by various ESMs driven by identical anthropogenic emissions is diverse and highly uncertain in the prediction of atmospheric CO2 concentration (Friedlingstein et al., 2006, 2014; Hoffman et al., 2013). Many previous studies (Friedlingstein et al., 2006; Hoffman et al., 2013; Anav et al., 2013; Aroa et al., 2013; Friedlingstein et al., 2014) suggested that the uncertainty of CO2 concentrations simulated by the emission-driven ESMs should be attributed to the carbon cycle over land rather than over ocean. In particular, one of the main. . .."

P3 L90-92 Make it clear you are talking about direct field characterization of global budgets for soil heterotrophic as opposed to indirect carbon budgeting estimates. Right now it reads like no one has ever looked at measuring soil respiration at all which is completely false as the authors go into detail later on.

We agree and modify the sentence as: (L90-92) "However, the amplitude of soil decomposition process has not been quantified through direct field measurements in the global domain, and highly uncertain, mostly due to the lack of observation data and

poor estimates of it indirectly from soil temperature (Sussela et al., 2012)."

P5 L119-121 You need a citation to back up this statement. I suggest (Todd-Brown et al., 2013) for a review of CMIP5 soil carbon models or directly citing the CMIP5 ESM manuscripts themselves.

As your comments, we include the relevant papers in the sentence: (L119-121) "However, most advanced ESMs partcipated in Coupled Model Intercomparison Project Phase 5 (CMIP5) still use a globally-constant Q10 value in the dynamic global vegetation models (Anav et al. 2013; Todd-Brown et al. 2013)."

P6 L157 Please make it clear if you are using the global soil map or underlying data set from (Hashimoto et al., 2015). Given the reference to regridding I'm assuming this is the soil map product (if this is not the case please clarify and disregard the following comments). This is a fatal flaw in this study. While the model used to generate this data product is not explicitly a Q10 model it is also clearly not in situ observations which makes this study a reanalysis of an existing model not a new interpretation of observations as the authors have framed this manuscript.

No, this study did not use the "global soil map" data but the gridded global map of soil respiration data from Hashimoto et al. (2015).

As this comment is same as in the above, please check our responses there (the first response in the major points).

P11 Sect 3.1 Why are we looking at GPP here? (Anav et al., 2013) Already looked at GPP in the context of FLUXNET, how is this different? This section still seems disconnected from the rest of the results as was mentioned in above comments on the abstract.

The original manuscript is lack of reasons why this study also examined the changes in GPP. As we answered in the above, we hypothesized that the improvement of soil respiration process by implementing variable Q10 in the model should also improve the

representation of GPP in the C-N (carbon-nitrogen) coupled ESMs.

The model intercomparison for GPP simulations by CMIP5 ESMs was to highlight the deficiencies in the GPP simulation by the C-N coupled models. The C-N coupled ESMs (i.e., CESM-BGC, NorESM) significantly overestimated (underestimated) GPP in the tropics (high latitude regions) compared with the rest of ESMs without C-N coupling. The impacts of new parameterization in this study on the GPP simulation is the reduction of systematic biases of GPP spatial distribution.

For a better connection, We reconstructed results part for single section from particulars section. And we revised manuscript as : (L271) "This study further compares the simulation of GPP by various ESMs in CMIP5."

P12 L293-296 This seems to belong in the GPP section. Unless you are also applying the Q10 sensitivity analysis to the GPP product in which case you need to be clearer how that ties into the methods section Eq 4-8 soil referred to Rs.

(L293-296) These sentences are redundant to the previous sections and will be removed in the revised manuscript.

P12 Please avoid the use of acronyms where possible. CTL and EXT break the flow of the manuscript. We use these acronyms for brevity. We will carefully revise the manuscript and improve flowing.

P20 L490 Malformated citation (bad first author name) We correct it as below:(L490).

References: Anav, A., Friedlingstein, P., Kidston, M., Bopp, L., Ciais, P., Cox, P., Jones, C., Jung, M., Myneni, R. and Zhu, Z.: Evaluating the land and ocean components of the global carbon cycle in the CMIP5 Earth system models, J. Clim., 26, 6801–6843, doi:10.1175/JCLI-D-12-00417.1, 2013. Arora, V. K., Boer, G. J., Friedlingstein, P., Eby, M., Jones, C. D., Christian, J. R., Bonan, G., Bopp, L., Brovkin, V., Cadule, P., Hajima, T., Ilyina, T., Lindsay, K., Tjiputra, J. F., Wu, T.: Carbon–concentration and carbon–climate feedbacks in CMIP5 earth system models, J. Clim., 26, 5289-5314,

doi:10.1175/JCLI-D-12-00494.1, 2013. Davidson, E. A. and Janssens, I. A.: Temperature sensitivity of soil carbon decomposition and feedbacks to climate change., Nature, 440, 165–173, doi:10.1038/nature04514, 2006. Davidson, E. A., Janssens, I. A. and Luo, Y.: On the variability of respiration in terrestrial ecosystems: moving beyond Q10, Glob. Chang. Biol., 12(2), 154–164, doi:10.1111/j.1365-2486.2005.01065.x, 2006. Friedlingstein, P., Cox, P., Betts, R., Bopp, L., von Bloh, W., Brovkin, V., Cadule, P., Doney, S., Eby, M., Fung, I., Bala, G., John, J., Jones, C., Joos, F., Kato, T., Kawamiya, M., Knorr, W., Lindsay, K., Matthews, H. D., Raddatz, T., Rayner, P., Reick, C., Roeckner, E., Schnitzler, K. G., Schnur, R., Strassmann, K., Weaver, A. J., Yoshikawa, C., and Zeng, N.: Climate–carbon cycle feedback analysis: Results from the C4MIP model intercomparison, J. Clim., 19, 3337–3353, doi:10.1175/JCLI3800.1, 2006. Friedlingstein, P., Meinshausen, M., Arora, V. K., Jones, C. D., Anav, A., Liddicoat, S. K., and Knutti, R.: Uncertainties in CMIP5 climate projections due to carbon cycle feedbacks, J. Clim., 27, 511-525, doi:10.1175/JCLI-D-12-00579.1, 2014. Hashimoto, S., Carvalhais, N., Ito, A., Migliavacca, M., Nishina, K. and Reichstein, M.: Global spatiotemporal distribution of soil respiration modeled using a global database, Biogeosciences, 12(13), 4121–4132, doi:10.5194/bg-12-4121-2015, 2015. Hoffman, F. M., Randerson, J. T., Arora, V. K., Bao, Q., Cadule, P., Ji, D., Jones, C. D., Kawamiya, M., Khatiwala, S., Lindsay, K., Obata, A., Shevliakova, E., Six, K. D., Tjiputra, J. F., Volodin, E. M., and Wu, T.: Causes and implications of persistent atmospheric carbon dioxide biases in Earth System Models, J. Geophys. Res. Biogeosci., 119, 141-162, doi: 10.1002/2013JG002381, 2013. Karhu, K., Auffret, M. D., Dungait, J. A. J., Hopkins, D. W., Prosser, J. I., Singh, B. K., Subke, J.-A., Wookey, P. A., Ågren, G. I., Sebastià, M.-T., Gouriveau, F., Bergkvist, G., Meir,P., Nottingham, A. T., Salinas, N. and Hartley, I. P.: Temperature sensitivity of soil respiration rates enhanced by microbial community response, Nature, 513(7516), 81–84, doi:10.1038/nature13604, 2014. Mahecha, M. D., Reichstein, M., Carvalhais, N., Lasslop, G., Lange, H., Seneviratne, S. I., Vargas, R., Ammann, C., Arain, M. A., Cescatti, A., Janssens, I. A., Migliavacca, M., Montagnani, L. and Richardson, A. D.:

Global convergence in the temperature sensitivity of respiration at ecosystem level, Science (80-. )., 329(5993), 838–840, doi:10.1126/science.1189587, 2010. Oleson, K., Lawrence, D. M., Bonan, G. B., Drewniak, B., Huang, M., Koven, C. D., Levis, S., Li, F., Riley, W. J., Subin, Z. M., Swenson, S. C., Thornton, P. E., Bozbiyik, A., Fisher, R., Heald, C. L., Kluzek, E., Lamarque, J.-F., Lawrence, P. J., Leung, L. R., Lipscomb, W., Muszala, S., Ricciuto, D. M., Sacks, W., Sun, Y., Tang, J., and Yang, Z.-L.: Technical Description of version 4.5 of the Community Land Model (CLM), NCAR Technical Note NCAR/TN-503+STR, Boulder, Colorado, 420 pp., 2013. Raich, J. W., Potter, C. S., and Bhagawati, D.: Interannual variability in global soil respiration, 1980–1994, Glob. Change Biol., 8, 800–812, 2002. Shao P., Zeng, X., Sakaguchi, K., Monson, R. K., and Zeng, X.: Terrestrial carbon cycle: climate relations in eight CMIP5 earth system models, J. Clim., 26, 8744-8764, doi:10.1175/JCLI-D-12-00831.1, 2013. Sheffield, J., Goteti, G., and Wood, E. F.: Development of a 50-Year High-Resolution Global Dataset of Meteorological Forcings for Land Surface Modeling, J. Clim., 19, 3088-3111 doi: http://dx.doi.org/10.1175/JCLI3790.1, 2006. Qi, Y., Xu, M., and Wu, J.: Temperature sensitivity of soil respiration and its effects on ecosystem carbon budget: nonlinearity begets surprises, Ecolog. Model., 153, 131–142, 2002 Todd-Brown, K. E. O., Randerson, J. T., Post, W. M., Hoffman, F. M., Tarnocai, C., Schuur, E. A. G. and Allison, S. D.: Causes of variation in soil carbon simulations from CMIP5 Earth system models and comparison with observations, Biogeosciences, 10(3), 1717–1736, doi:10.5194/bg-10-1717-2013, 2013. Xu, M., and Qi, Y.: Spatial and seasonal variations of Q10 determined by soil respiration measures at a Sierra Nevadan forest, Global Biogeochem. Cy., 15, 687 – 696, 2001. Zhou, T., Shi, P., Hui, D., and Luo, Y.: Global pattern of temperature sensitivity of soil heterotrophic respiration (Q10) and its implications for carbon-climate feedback, J. Geophys. Res., 114, doi:10.1029/2008JG000850, 2009.

Please also note the supplement to this comment:
http://www.biogeosciences-discuss.net/bg-2016-549/bg-2016-549-AC2-

[Figure]

supplement.zip

[Figure]

[Figure]

**Fig. 1.** Figure S1. Spatial distribution of turnover time (year) of soil carbon in (a) CTL and (b) EXP. (c) indicates the difference between EXP and CTL simulation. The turnover time is defined as the ratio o

[Figure]

---

## Author Comment (AC3) · 7 May 2017

Anonymous Referee #2 Q10 is a critical parameter for simulating soil respiration and C cycle and therefore feedback between C cycle and climate. Yet most ESMs adopt constant Q10, which can possibly lead to unrealistic simulations of soil processes and other C cycle processes. Therefore, this paper contributes to improve model performance by implementing a new Q10 parametrization. This has significant implications to modeling studies.

We appreciated the reviewer's thorough and constructive comments. Below is our

[Figure]

point-by-point response to the specific comments.

I have two major concerns. One is the authors did not explain why improvement of Q10 parameterization and resulting improvement of soil respiration can help improve simulations in GPP. Is it because nitrogen availability resulting from changed soil respiration rates or other mechanisms?

This study implemented the variable Q10 in the parameterization of soil decomposition flux, which directly affects the heterotrophic respiration from soil organic matter (SOM). In addition, the CLM4 model used in this study has the interactive carbon-nitrogen (C-N) cycle, by which it changes the plant assimilation and GPP in the meantime. For example, a higher (lower) Q10 value induces a faster (slower) carbon decomposition rate in the model, and it tends to increase (decrease) nitrogen assimilation from soil to vegetation, thereby increasing (decreasing) GPP by plants.

In CLM4 (Olsen et al., 2013), plant nitrogen uptake from soil mineral nitrogen pool is separated by plant demand for mineral nitrogen from the soil (NFplant_demand_soil) and retranslocated nitrogen (NFretrans) which construct to mobilize senescing tissues. Therefore, total plant nitrogen uptake from soil mineral nitrogen pool is :

$$\text{ãĂŰNFãĂŮ\_(plant\_demand\_soil)}= \text{ãĂŰNFãĂŮ\_(plant\_demand)}-\text{ãĂŰNFãĂŮ\_retrans}$$

This total plant nitrogen demand for new growth (NFplant_demand) is calculated by total carbon available for new vegetation growth allocation (CFavail_alloc) from soil as:

$$\text{ãĂŰNFãĂŮ\_(plant\_demand)} \ \text{ãĂŰCFãĂŮ\_(avail\_alloc)} \ N\_allom/C\_allom$$

where CFavail_alloc is related with carbon amount in each carbon pools. These processes induce that more carbon decomposition enhanced more nitrogen supplement from soil to plant for new plant growth (increasing GPP).

This aspect is discussed in the introduction section as below: (L130) "Realistic spatial distribution of soil decomposition processes affect not only Rs but also primary production by improving nitrogen assimilation from soil to vegetation"

[Figure]

We further examined the impact of Rs on plants net primary production. The variable Q10 parameterization tends to affect the turnover time of soil carbon, which is defined as the soil carbon amount divided by net primary production (NPP) (i.e., soil carbon/NPP). As shown in Fig. S1 below, the run with variable Q10 (EXP) makes shorter turnover time in northern hemisphere high latitudes and longer in the tropics compared with the control run(CTL). The shorter turnover time in high latitudes suggests the enhancement of nitrogen assimilation to vegetation in EXP, thereby enhancing net primary production by plants. We will address this change in the manuscript as below:

(L336) "The variable Q10 in the parameterization of soil decomposition flux immediately affects the heterotrophic respiration from soil organic matter (SOM) as given by the model formulations in Eq. (1) and (2). Moreover, this modification changes the plant assimilation and GPP in the meantime in this carbon-nitrogen coupled model. A faster (slower) carbon decomposition rate in the model tends to increase (decrease) nitrogen assimilation from soil to vegetation and plants, thereby increasing (decreasing) GPP. This aspect is illustrated well by comparing the turnover time of the soil carbon, which is defined as the ratio of soil carbon amount to the net primary production (NPP), between the CTL and EXP runs (See Fig. S1 below; Fig. 8 in the revised manuscript). As shown in the figure, the run with variable Q10 (EXP) makes shorter turnover time in northern hemisphere high latitudes and longer in the tropics compared with the control run(CTL). The shorter turnover time in high latitudes suggests the enhancement of nitrogen assimilation to vegetation in EXP, thereby enhancing net primary production by plants."

Another is the confusion of sensitivity of soil respiration to temperature (i.e., Q10) and sensitivity of Q10 to parametrization.

From the Eq. (2), the soil respiration in this model is already an exponential function of soil temperature. The Q10 value in the original scheme is a global constant of 1.4, and therefore, the sensitivity of soil respiration to temperature is constant, regardless

of plant functional types. This study implemented a state-dependent Q10 parameterization for each 17 plant functional types. As we discussed in the text (L216-219), "the dependence of Rs on soil moisture and temperature can be dependent on PFT", and "this approach is to consider the nonlinear relationship between Rs and the two major soil environmental factors (i.e., soil temperature and moisture [Davidson et al., 1998; Raich et al., 2002]".

Specific comments are listed below.

Lines 84-85. Please cite literature(s) for this statement.

Todd-Brown et al. (2013) emphasized the importance of soil carbon pool in the carbon exchange between atmosphere and land.

(Addition after L85) "(Todd-Brown et al., 2013)".

Lines 89-90. There are many studies on soil respiration under experimental warming that which should address this point better.

We will add more reference in the revised manuscript as: "Many studies investigated the response of soil respiration (Rs) under global warming, and most of them suggested the warming would accelerate the release of CO2 from soil in future (Cox et al., 2000; Dufresne et al., 2002; Friedlingstein et al, 2003; Suseela et al., 2012)."

Lines 93-94. There is a recent review paper talking about this issue (Global Biogeochemical Cycles, 2016, 30: 40-56)

We will add this recent review paper: (After L93)"Moreover, Luo et al., (2016) suggested that optimizing parameters in the current ESMs are needed based on observations for improving soil carbon projection in the models. The reduction of uncertainty in the parameterizations of the biogeochemical process in the soil system remains a challenge for the ESM modeling community."

Lines 102-103. I would not say that because there are many field studies examined

Q10, but it is true, we lack long-term observation data on how Q10 changes overtime. It may be difficult for regular field studies to explore dependency of Q10 on temperature as they derive Q10 through entire seasonal temperature range when Rs is measured. However, it can be tested via lab incubation experiments or field experiments with warming treatments.

Following the reviewer's comment, we will delete this sentence (L102-103).

Lines 104-105. I think they meant Rs tends to decrease with temperature increase. This needs to be clearly stated.

Belay-Tedla et al. (2009) suggested that warming-induced changes in plant growth and community structure can considerably influence the quality and quantity of substrates which in turn regulates the responses of soil respiratory C release to rising temperature.

We changed this sentence (L104-105).

Line 145. Add "time resolution/interval/step" before "is monthly".

We will add the time resolution for the data in this sentence. (L145)

Line 151. "Each tile is 1200 X1200 km" needs to be clearer. Does it refer to original MODIS17A3 GPP and NPP data?

We will remove this sentence.

Lines 155-156. Which year are these data for or are they the average from 2000 to 2006?

It is based on the average of 2000 – 2006.

Line 157. This sentence is not clear. Did they mean that they compared modeled Rs against the data by Hashimoto et al. (2015) for validating their model? What is time period of Rs data by Hashimoto et al. (2015)?

(L157) The sentence will be clarified as:

"Simulations of soil respiration (Rs) by CLM4 will be verified using the gridded reanalysis dataset from Hashimoto et al. (2015), which has the data period of 1983-2005."

Line 159. What is "SRDB"? It needs to be fully spelt.

We add the full name for SRDB (L159) : "soil respiration database (SRDB) version 3 (Bond-Lamberty and Thomson, 2010)"

Lines 165-167. What does "assuming" mean?

Hashimoto et al. (2015) developed the semi-empirical model parameterized with many Rs data points using near surface temperature and precipitation. Using this semi-empirical model, Hashimoto et al. (2015) derive the long-term gridded Rs data.

(L167) "Assuming" will be replaced with "deriving".

Lines 176-177. Why is the decomposition flux calculated by multiplying carbon amount from dead leaf? Soil respiration include both microbial respiration and root respiration. The substrate for microbial respiration is SOM in the soil, which is originally derived from litter (dead leaf, wood, and root). And root respiration is respiration by live roots, which is related to root biomass. If their model does not separate the two components, it should be carbon content of soil.

This is our mistake and it is supposed to be "litter". In CLM4, the root respiration is from live roots as your comments. Therefore, the heterotrophic respiration of soil is from microbial respiration from SOM. (L176-177) "dead leaf" → "litters"

Lines 186-187. It should be "the water potential for soil decomposition". We modified this sentence.(L186-L187)

Lines 212-213. In a multiple regression analysis, why are the relationships between Rs and T and between Rs and M separately?

In our parameterization, Q10 is changed not only by temperature but also by soil moisture. This is same as in Qi et al. (2002).

Using Eq. (8), multiple regression analyses were conducted for each plant functional types simultaneously with soil temperature and moisture.

Lines 224-235. This paragraph is very confusing. The authors need to give time step of each dataset to avoid confusion. Are GPCP and TRMM data used to rescale precipitation data by Sheffield et al. (2006) to a time step of daily and 3-hour step for forcing CLM? And CLM is forced by 3hr data, then why daily data are needed? Is it for regression analysis? In addition, Sheffield et al. (2006) data have radiation, so what are radiation data by NASA for? If they are for all descriptions of Sheffield et al. (2006), it needs to be clear.

The whole paragraph is revised as: (L224-237) "To obtain these variables, this study conducted the land surface reanalysis for recent 30 years (1981 – 2010), using the off-line land surface model driven by observed meteorological forcing data archived by Sheffield et al. [2006]. The 3-hourly forcing data by Sheffield et al. (2006) consists of the National Centers for Environmental Prediction–National Center for Atmospheric Research reanalysis (Kalnay et al., 1996), which were corrected with independent observations. For precipitation, the daily Global Precipitation Climatology Project (GPCP, Huffman et al., 2001) data were processed into the 3-hourly data using the Tropical Rainfall Measuring Mission (TRMM: Huffman et al., 2003) 3B42RT but constraining daily mean amount from GPCP. Surface temperature was constrained by the observation from the monthly Climatic Research Unit (CRU) 2.0 product (Mitchell et al., 2004). The observed radiation was also used from the monthly NASA Langley surface radiation budget (Stackhouse et al., 2004) data. Remaining meteorological conditions such as surface wind and humidity were from the National Centers for Environmental Prediction–National Center for Atmospheric Research (NCEP-NCAR) atmospheric reanalysis. Interested readers refer to Sheffield et al. (2006) for the detail. Using this 3-hourly forcing data, this study integrated the off-line land surface model with 3-hourly time steps and at a $0.5° \times 0.5°$ spatial resolution."

Line 226. Year for the literature is needed.

We will modify as "The forcing data archived by Sheffield et al. (2006). . ..".

Lines 236-237. How can an audience find a reference that is not published?

We delete this sentence in the revised manuscript.

Line 238. I think they are talking about soil respiration by Hashimoto et al. (2015), but this needs to be specified.

It is correct. We specified the data sources as (Modification from L238~) : "Figure 1 shows the r-squared values from the multiple regression analysis between the soil respiration data from Hashimoto et al. (2015) and the soil temperature and moisture derived from the off-line land surface model. The result is presented for each PFT type in the figure.

Lines 246-247. The period of forcing data by Sheffield et al. (2006) is reduced to be the same with GSWP2?

Yes, and when we adjust the sampling period from the long-term period (1983-2010) to the short one (1986-1995), the r-squared values become similar as the ones obtained using GSWP2 data for the period of 1986-1995. This suggests the sampling period for regression is important (See Figure S2 below). This study conducted the offline land surface model integration forced by observational forcing for a sufficiently long period (e.g., 1983-2010), which is to better represent the subsurface climatology at the presence of strong interannual variability.

Line 251. Does this mean a global constant and unchanged over time? Fig. 1. Why 28 years? Is this because of time period of data by Hashimoto et al. (2015)? The title is confusing and needs to be revised. It would be useful for audiences to see regression models for all PFT as supplementary information.

The CTL simulation is global constant of Q10 and unchanged over time. In contrast, the EXP changed the Q10 in every time step in the model at the change of soil temperature and moisture. The relationship is taken empirically from the multiple regression

analysis. To remove the impacts from interannual variability in meteorological data due to El Nino/La Nina cycles, this study calibrated the regression model for the long period of 1983-2010, which is the longest available period for the Hashimoto et al.'s data.

We attach the Excel file for the regression analysis for each 17 PFTs.

Line 252. CESM run needs to be described before "Figure 2".

This CESM data was obtained from CMIP5 ESMs results. We indicate this in the manuscript: (L252) "The offline simulations for GPP and soil respiration are also compared with those from the fully-interactive Community Earth System Model with Biogeochemistry (CESM-BGC) model simulation that used the identical land surface model (i.e., CLM4). The dataset was obtained from Earth System Grid – Center for Enabling Technologies (ESG-CET at http://pcmdi9.llnl.gov/)."

Fig. 3. CTL should be included in the figure title.

We will modify from "CLM4" to "CTL" in Figs. 3c and 3f.

Line 272. The title should be "results and discussions".

We will change the section title as suggested.

Line 330. Sensitivity of Rs to soil temperature?

We will change it to "Sensitivity of Rs and soil temperature" to "sensitivity of Rs to soil temperature (L330)".

Line 334. Which panel in Fig. 6 did they refer to by "enhanced relationship between Rs and temperature" for the northern Eurasian and Chinese regions?

It is possible to misunderstand this sentence. The Eurasian and Chinese regions is mostly covered boreal shrub land and crop field. The regions which are enhanced relationship between Rs and soil temperature are improved to simulate GPP.

Line 273 and the whole paragraph. In method section they never talked about CMIP5,

why here a subtitle for CMIP5 GPP? If they use CMIP5 to evaluate or compare CLM EXP, they needs to describe it in method and need to include EXP or CTL in this section and in Fig. 4. They also need to give some details such as how many CMIP5 models, model names and if MME includes CESM-BGC and NorESM. There are few papers with figures that do not include results from their present study. If they want to discuss the issue of underestimation GPP by coupled N cycle, this part should be put into discussion and the figure should be in the supplementary document. Overall, this part is not very relevant to the main purpose of this study.

This study used the GPP data simulated by 10 emission-driven CMIP5 ESMs. We add the list of ESMs in Table 2 in the revised manuscript.

The original manuscript is lack of reasons why this study also examined the changes in GPP. As we answered in the above, we hypothesized that the improvement of soil respiration process by implementing variable Q10 in the model should also improve the representation of GPP in the C-N (carbon-nitrogen) coupled ESMs.

The model intercomparison for GPP simulations by CMIP5 ESMs was to highlight the deficiencies in the GPP simulation by the C-N coupled models. The C-N coupled ESMs (i.e., CESM-BGC, NorESM) significantly overestimated (underestimated) GPP in the tropics (high latitude regions) compared with the rest of ESMs without C-N coupling. The impacts of new parameterization in this study on the GPP simulation is the reduction of systematic biases of GPP spatial distribution.

For a better connection, We reconstructed results part for single section from particulars section. And we revised manuscript as : (L271) "This study further compares the simulation of GPP by various ESMs in CMIP5."

Figure 4. Figure title needs more information. "MME" needs to be fully spelt. I would use "CMIP5" instead of "MME" in the legend. Are blue bars the average of CESM-BGC and NorESM? Why no results from EXP? Global GPP can also be shown in this figure since it is mentioned in the text. In addition, no y axis (unit) in this figure.

We add the Figure title and the label for y axis in Fig. 4. "MME" is replaced with "CMIP5" as suggested. Also detailed information for blue bars are added in the caption.

As this paragraph is for the overall simulation bias of GPP by CMIP5 ESMs form the interactive climate-carbon feedback simulations, we do not include the offline simulation results from CTL and EXP. Instead, those are given for CTL and EXP in Fig. 8 in the original manuscript.

Line 288. What are they talking about by "these two"? In addition, according to the figure, GPP 60N-80N is not major region.

We delete the sentence for clarification, and modify the preceding sentence as: (L287-290) "These systematic biases in the tropics and the Northern Hemisphere high latitudes are common in the C-N coupled models based on CLM4 (Bonan et al. 2009)."

Lines 293-296. Delete this since it is a repeat of last paragraph.

We delete the sentences in the revised manuscript.

Lines 296-300. How could they conclude that from Fig. 5 since a) is difference between EXP and observation, not the absolute values of observation and EXP. They can show maps of all three data sets (observation, EXP and CTL) in supplementary documents to support this statement.

As we already show the spatial pattern of Rs from Hashimoto et al. and the offline simulation (CTL) in Fig. 3d and 3f, respectively, we only show the difference pattern of EXP minus observation in Fig. 5a. We add the supplementary figure in the manuscript below for the discussion.

Fig. 5. I would suggest to add another panel for the difference between CTL and observation. Unit is missing for both panels.

This is already given in Fig. 3f. We add the unit and Y axis in the figure and caption. Also, we changed caption in figure 3 for avoiding confusing.

Lines 293-331. It would be easier to give panel number such as "Fig 5a)". Thank you
for your comment. We will add the specific figure title in this paragraph.

Lines 312-313. Fig. 6 does not support this point because the y axis is the difference
between EXP and CTL. It is not the absolute Rs of EXP or CTL. More changes do
not necessarily result in higher absolute Rs. The point may be supported if they draw
scatter plots for both EXP and CTL in each panel and show better correlation between
Rs and temperature in EXP than CTL. Line 314. "The difference between EXP and
CTL increases with temperature" is not supported by Fig. 6 since they are only the
cases in a few panels

The relationship between absolute Rs and soil temperature is not much different be-
tween CTL and EXP simulation. In figure S4, we provided the r-squared value between
log Rs and soil temperature. All vegetation types are positive relationship between log
Rs and soil temperature in both simulation.

Temperate, tropical forest and Grass regions show relative higher positive relationship
between Rs and temperature in CTL simulation comparing with EXP simulation. How-
ever, EXP simulation has higher positive r-squared value in cold temperature regions
(e. g., Boreal forest and Boreal shrub). Therefore, non-uniform Q10 value affects more
in cold regions than warm regions. Interestingly, PFTs of higher value of climatology
averaged Q10 value (Table 1 in manuscript) comparing with standard value (1.5) en-
hanced relationship between Rs and soil temperature such as boreal forest and boreal
shrubs except for crop land.

We added figure S4(Table 3 in manuscript) and modified manuscript as (L312-322):
"The sensitivity of Q10 parametrization depend on the surface vegetation types. For
instance, boreal forest and shrub regions which has cold climate shows significant en-
hanced relationship between Rs and soil temperature. In contrast, temperate, tropical
forest and Grass regions show relative higher positive relationship between Rs and
temperature in CTL simulation comparing with EXP simulation (Table 3). Some regions in shrub and crop regions is unclear to show this relationship. Interestingly, PFTs of higher value of climatology averaged Q10 value (Table 1) comparing with standard value (1.5) enhanced relationship between Rs and soil temperature such as boreal forest and boreal shrubs except for crop land."

Lines 319-322. This explanation is not convincing because tropical is the opposite and it cannot explain shrub, grass and crop.

We added the supplement figure and discussion in manuscript (L312-322) : "In boreal forest and shrub regions which has higher Q10 value comparing with global constant, the relationship between Rs and temperature in EXP are enhanced than CTL simulation. However, the tropical, temperate forests and grass regions (lower Q10 value than 1.5 in EXP simulation) is unclear for impacts of Q10 parameterization. One of possibility is that these regions are strong sensitivity of soil respiration to soil moisture. Figure S5 supported that this mechanism. In high temperature region, the sensitivity of Rs on the moisture is stronger than temperature. It reflected the unclear change of temperature sensitivity of Rs to soil temperature over tropical forest region."

Fig. 7. Unit should be given.

We add the unit in Fig. 7.

Lines 336-334. Please explain the mechanism for this.

Our parameterization modified the decomposition rate in the soil layers. As we responded in the above (in the Major Points), the variable Q10 values in space and time affects the heterotrophic respiration from soil organic matter (SOM). From the Eq. (1) and (2), in Sect. 2.2, a higher Q10 value tends to increase carbon decomposition into soil layers. Enough nutrients in soil layers induces more carbon assimilation to vegetation and plants. This impact on GPP is reflected on the simulated turnover time difference between CTL and EXP (Please check Fig. S1 above). The results suggest that Q10 variation influences on gross primary production (GPP), which response

depends on region and different in the tropics and the high latitudes.

Lines 337-340. The global GPP in FLUXNET should also be given here. "SH" should be fully spelt.

(L340) "SH" is replaced with "Southern Hemisphere".

Line 344. Are the numbers in Fig. 8 zonal mean or zonal sum? I think they are sums.

This is zonal mean of GPP.

Line 345. The word "budget" is not suitable here. Use GPP.

GPP is more suitable in that sentence.

Line 348. Are they talking about Fig. 3? Please indicate.

It indicates Figure 4 (zonal averaged GPP in CMIP5-ESMs and C-N coupled model). We will add the figure number at the end of sentence (L348).

Fig. 8. Adding global data to this figure would help. This figure should merged with Fig. 7 (i.e., three panels). Y axis is missing.

We add the global average of GPP in Fig. 8 in the revised manuscript. We also add the label and unit for y axis.

Lines 349-350.What did the authors mean here? How can carbon pool in the soil system affect plant assimilation? Plants do not absorb carbon in soil.

Nitrogen decomposition is closely related with carbon decomposition in CLM4. We modify it as: "The modification to the soil process parameterization can affect the rest of the terrestrial carbon cycle by changing the carbon pools and nitrogen pools in the soil system needed for plant nitrogen assimilation"

Fig. 9. No y axis.

We will add the Y axis.

Lines 398-400. Please cite literature here.

We add the literatures (L400, Bonan et al. 2010 and Bonan et al., 2011).

Lines 404-405. This sentence is not clear.

We will modify this sentence. (L404) : "In fact, the parameterization of photosynthesis in the state-of-the-art ESMs is implemented in a similar fashion with small differences, based on the formulations from Farquhar et al. (1980). "

References: Bonan, G. B., Lawrence, P. J., Oleson, K. W., Levis, S., Jung, M., Reichstein, M., Lawrence, D. M., and Swenson, S. C.: Improving canopy processes in the Community Land Model version 4 (CLM4) using global flux fields empirically inferred from FLUXNET data, J. Geophys. Res., 116, G02014, doi:10.1029/2010JG001593, 2011. Bonan, G. B., and Levis S.: Quantifying carbon‐nitrogen feedbacks in the Community Land Model (CLM4), Geophys. Res. Lett., 37, L07401, doi:10.1029/2010GL042430, 2010. Cox, P. M., Betts, R. A., Jones, C. D., Spall, S. A., Totterdell, I. J., :Acceleration of global warming due to carbon-cycle feedbacks in a coupled climate model, Nature, 408, 184-187, 2000. Dufresne J. L., Friedlingstein, P., Berthelot, M., Bopp, L., Ciais, P., Fairhead, L., Le Treut, H., Monfray, P.: On the magnitude of positive feedback between future climate change and the carbon cycle, Geophys. Res. Lett., 29, 1405, 43 , 2002. Farquhar G. D., Caemmerer, S., Berry, J. A..:A biochemical model of photosynthetic CO2 assimilation in leaves of C3 species, Planta, 149, 78–90, 1980. Friedlingstein P., Dufresne, J. L., Cox, P. M., Rayner, P.: How positive is the feedback between climate change and the carbon cycle?, Tellus-B, 55, 692–700,2003. Hashimoto, S., Carvalhais, N., Ito, A., Migliavacca, M., Nishina, K. and Reichstein, M.: Global spatiotemporal distribution of soil respiration modeled using a global database, Biogeosciences, 12(13), 4121–4132, doi:10.5194/bg-12-4121-2015, 2015. Luo, Y., Wan, S., Hui, D., and Wallace, L. L.: Acclimatization of soil respiration to warming in a tall grass prairie, Nature, 413, 622-625, doi:10.1038/35098065, 2001. Qi, Y., Xu, M., and Wu, J.: Temperature sensitivity of soil respiration and its effects

on ecosystem carbon budget: nonlinearity begets surprises, Ecolog. Model., 153, 131–142, 2002. Todd-Brown, K. E. O., Randerson, J. T., Post, W. M., Hoffman, F. M., Tarnocai, C., Schuur, E. A. G. and Allison, S. D.: Causes of variation in soil carbon simulations from CMIP5 Earth system models and comparison with observations, Biogeosciences, 10(3), 1717–1736, doi:10.5194/bg-10-1717-2013, 2013. Suseela, V., Conant, R. T., Wallenstein, M. D., and Dukes, J. S.: Effects of soil moisture on the temperature sensitivity of heterotrophic respiration vary seasonally in an old-field climate change experiment, Global Change Biol., 18, 336–348, 2012. Sheffield, J., Goteti, G., and Wood, E. F.: Development of a 50-Year High-Resolution Global Dataset of Meteorological Forcings for Land Surface Modeling, J. Clim., 19, 3088-3111 doi: http://dx.doi.org/10.1175/JCLI3790.1, 2006. Xu, M., and Qi, Y.: Spatial and seasonal variations of Q10 determined by soil respiration measures at a Sierra Nevadan forest, Global Biogeochem. Cy., 15, 687 – 696, 2001. Zhou, T., Shi, P., Hui, D., and Luo, Y.: Global pattern of temperature sensitivity of soil heterotrophic respiration (Q10) and its implications for carbon-climate feedback, J. Geophys. Res., 114, doi:10.1029/2008JG000850, 2009.

Please also note the supplement to this comment:
http://www.biogeosciences-discuss.net/bg-2016-549/bg-2016-549-AC3-supplement.zip
* * *
[Figure]

**Fig. 1.** Figure S1. Spatial distribution of turnover time (year) of soil carbon in (a) CTL and (b) EXP. (c) indicates the difference between EXP and CTL simulation. The turnover time is defined as the ratio

[Figure]

**Fig. 2.** Figure S2. R-squared values for each PFT from the multiple regression analysis between Hashimoto et al.'s soil respiration data and the three different meteorological datasets for soil temperature and

[Figure]

**Fig. 3.** Figure S3. (a) The spatial distribution of Rs from Hashimoto et al. (2015) and bias pattern of Rs in (b) CTL simulation and (c) EXP simulation.

|      | Temperate | Boreal | Tropical | Shrub | B. Shrub | Grass | Crop |
|------|-----------|--------|----------|-------|----------|-------|------|
| CTL  | 0.40      | 0.06   | 0.34     | 0.06  | 0.04     | 0.38  | 0.27 |
| EXP  | 0.36      | 0.25   | 0.31     | 0.05  | 0.20     | 0.28  | 0.31 |

**Fig. 4.** Figure S4. R-squared values between log Rs and soil temperature by PFTs in CTL and EXP

**Fig. 5.** Figure S5. Spatial Matrix between soil moisture (Y-axis) and soil temperature (X-axis) with Rs (color dots) in the CTL simulation.

---

## Author Comment (AC4) · 7 May 2017

Anonymous Referee #3 The authors developed PFT-dependent $Q_{10}$ values for soil organic matter (SOM) decomposition processes using a multiple regression method. They demonstrated that the spatially-distributed $Q_{10}$ had the potential to improve the simulation of both soil respiration and GPP compared with the CLM4 simulation with a uniform $Q_{10}$. It's necessary and important to use spatially-distributed $Q_{10}$ rather than a constant $Q_{10}$ in global simulations. I would like the authors to further clarify the "multiple regression" method used in this

study as I don't quite understand it while reading the manuscript:

We appreciate the reviewer's critical comments. We reply to the reviewer's comments in detail as in the below. For clarification, we also attach the Excel file showing data and the regression results between Rs and soil temperature and moisture for each 17 PFTs.

(1) what are the response variables (Rs?) and explanatory variables (T & M?) in the regression analysis?

The response variable is soil respiration (Rs) data from Hashimoto et al. (2015) and explanatory variables are soil temperature (T) and moisture (M) from off-line land surface reanalysis conducted in this study by driving land surface model with observation forcing from Sheffield et al. (2006).

(2) what datasets at what time-scale are used for regression?

This study used the monthly data for regression for the period of 30 years (1981-2010, total 360 samples each grid points).

(3) what is the relation between the equations 4-8 and the regression analysis?

The Q10 value in this study is defined as in Eq. (5) to (6). To solve it, this study defined "q" as in Eq. (4), which represents the fractional change of Rs. This is determined by subsurface soil temperature and other abiotic factors such as moisture, which is represented by Eq. (7)-(8).

From Eq. (8), if we know the change rate of Rs by soil moisture $((\partial R\_s (T,M))/\partial M)$ and temperature $((\partial R\_s (T,M))/\partial T)$, we can obtain q(T,M) and Q10 subsequently. Multiple regression analysis was conducted for each plant functional type (PFT) for solving q in Eq. (9).

(4) how do you calculate Q10 at every time interval as you stated in Line 381? Q10 is temperature-dependent as indicated in Eqs. 2 & 5, do you mean that you will also
change Q10 based on the temperature at current time-step? Another concern of mine is related to the calculation of Q10 using soil respiration data. We know that generally soil respiration includes both heterotrophic respiration from SOM decomposition and root respiration (growth + maintenance). It seems the PFT-dependent Q10 is developed for SOM decomposition processes, thus how do you use total soil reparation to determine the Q10 for SOM decomposition?

In the original CLM4 model used in this study, the decomposition rate is calculated in every time step using Eq. (1)-(3). During this calculation, the Q10 value is also needed to be determined for Eq. (2). Soil temperature (T) and moisture (M) for the current time step for the top 5 soil layers are used in the calculation.

As the reviewer's comment, soil respiration (Rs) consists of root respiration (root autotrophic respiration; Ra) and heterotrophic respiration (Rh). Rh of soil organic matter (SOM) is closely dependent on Q10 variation. Rs data from Hashimoto et al. (2015) does not separate explicitly Ra and Rh. Instead, they used the empirical relationship between Rs and Rh derived from the meta-analysis (Bond-Lamberty et al., 2014; See Seection 2.5 in Hashimoto et al., 2015). Ra is the residual from Rs minus Rh. The relationship which they adopted is as following:

lnâĄą(R_h )=1.22+0.73 lnâĄąãĂŰ(R_s)ãĂŮ

Therefore, we used total Rs data to determine Rh of SOM in this study.

Minor comments:

(1) Fig.5 & Fig. 9: please indicate the units of Rs and Ra. In addition, please explain what are Ra and Rs, i.e., plant autotrophic respiration and soil respiration.

We added the units and the definition of Ra and Rs clearly in the revised manuscript.

(2) Figs.4, 7, 8 & 9: please indicate the units of GPP.

We added the units.

(3) Line 304: "The Rs Simulation difference between CTL and EXP is given in Figure 5, in terms of global distribution as well as zonally-averaged distribution". I understand we may identify the zonal difference between CTL and EXP. However, Fig.5a shows the difference between EXP and Hashimoto data, not between EXP and CTL.

We modified the sentence as:.

(L304-305) The changes in Rs simulation by EXP are given in Fig. 5, in terms of global distribution as well as zonally-averaged distribution.

(4) Line 314: "the difference between EXP and CTL increases with temperature". It may be true for boreal and B_Shrub PFTs. I would suggest doing statistical tests to show whether the relation is significant or not.

Following the reviewer's comment, we conducted the significant test for the changes.

Figure 6 shows the scatter plots for Rs change (EXP minus CTL in y-axis), as a function of soil temperature (x-axis). Presumably due to the sub-biome variability in biotic and abiotic conditions, the scatter plots exhibit some nonlinearity in the curvature, but most of the values lie in the positive range for the moderate to warm temperatures (i.e., increased Rs at the given temperature by the variable Q10 formulation). Note that this relationship is not uniform in space, as the change of Q10 is not uniform in EXP (as shown in Fig. 2).

After adjusting 90% significant level in the scatter plots, the grid of grass, crop and shurbland are too many reduced. However, boreal, temperate forest regions and boreal shrub regions shows enhanced strong relationship between changed Rs and soil temperature in EXP comparing with CTL.

References Bond-Lamberty, B., and Thomson, A.: Temperature-associated increases in the global soil respiration record., Nature, 464, 579-582, doi:10.1038/nature08930, 2010. Hashimoto, S., Carvalhais, N., Ito, A., Migliavacca, M., Nishina, K. and Reichstein, M.: Global spatiotemporal distribution of soil respiration modeled using a global

database, Biogeosciences, 12(13), 4121–4132, doi:10.5194/bg-12-4121-2015, 2015.
Qi, Y., Xu, M., and Wu, J.: Temperature sensitivity of soil respiration and its effects on ecosystem carbon budget: nonlinearity begets surprises, Ecolog. Model., 153, 131–142, 2002.

Please also note the supplement to this comment:
http://www.biogeosciences-discuss.net/bg-2016-549/bg-2016-549-AC4-supplement.zip
* * *
[Figure]

**Fig. 1.** Figure S1. Scatter plots of change of Rs (y-axis) between EXP and CTL simulation as a function of soil temperature (x-axis). Scatters are calculated only over 90% significant level. Each panel shows